# XAttnMark: Learning Robust Audio Watermarking with Cross-Attention

Yixin Liu [1 2]   Lie Lu [2]   Jihui Jin [2]   Lichao Sun [1]   Andrea Fanelli [2]

## Abstract

The rapid proliferation of generative audio synthesis and editing technologies has raised serious concerns about copyright infringement, data provenance, and the spread of misinformation via deepfake audio. Watermarking offers a proactive solution by embedding imperceptible yet identifiable and traceable signals into audio content. While recent neural network-based watermarking methods like WavMark and AudioSeal have improved robustness and quality, they struggle to jointly optimize both robust detection and accurate attribution. This paper introduces Cross-Attention Robust Audio Watermark (XATTNMARK), which bridges this gap by leveraging partial parameter sharing between the generator and the detector, a cross-attention mechanism for efficient message retrieval, and a temporal conditioning module for improved message distribution. Additionally, we propose a psychoacoustic-aligned time-frequency (TF) masking loss that captures fine-grained auditory masking effects, improving watermark imperceptibility. XATTNMARK achieves state-of-the-art performance in both detection and attribution, demonstrating superior robustness against a wide range of audio transformations, including challenging generative editing at varying strengths. This work advances audio watermarking for protecting intellectual property and ensuring authenticity in the era of generative AI.

## 1. Introduction

With the rapid development of generative audio synthesis and editing techniques, anyone can now easily edit and synthesize audio content (OpenAI, 2024; Li et al., 2024; Copet

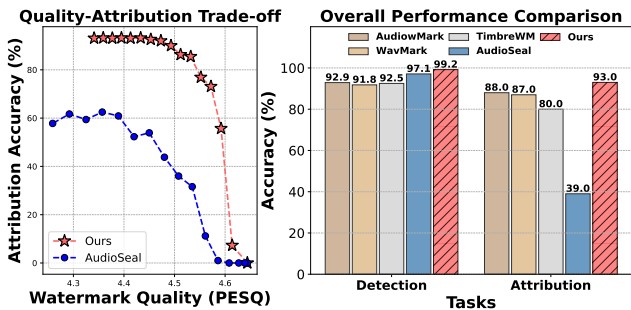

*Figure 1.* Quality-attribution performance trade-off curve across different watermarking strengths and the overall performance comparison on detection and attribution tasks. Higher values on both axes indicate better performance.

et al., 2024; Cao et al., 2025; Zhou et al., 2024). While it democratizes the creative process and enables new applications, it also raises serious concerns regarding unauthorized use of copyrighted data, data provenance, and authenticity (Pan et al., 2023; Shoaib et al., 2023; Park et al., 2023; Liu et al., 2024c; Meerza et al., 2025; Liu et al., 2024e). A notable example is the recent surge in deepfake audio and video, where malicious actors use generative techniques to impersonate and create fake speech and video content of online politicians or public figures, to spread misinformation and manipulate public opinion (Verma et al., 2024; Wenger et al., 2021; Buo, 2020; Bilika et al., 2023). Furthermore, beyond deepfake threats, the unauthorized exploitation of copyrighted content is also a growing concern in the AI industry (Singer, 2024; Qiwei et al., 2024; Brigham et al., 2024; Cui et al., 2024; Guan et al., 2024; Liu et al., 2024d). Nowadays many content creators are subject to copyright infringement due to the unauthorized use of their content for AI training and editing (Office, 2023; Abbott & Rothman, 2023). Original content is increasingly exploited and modified at scale, making it difficult to track data provenance (Cho, 2024; Robinson, 2024; Vermillio, 2024). As highlighted in Content Authenticity Initiative (CAI) (Wikipedia contributors, 2025), the challenge of combating disinformation and verifying content authenticity in the digital age requires robust standards for provenance metadata. Among the various solutions to track audio provenance and guarantee artists' protection (Ren et al., 2024; Desai & Riedl, 2024), watermarking stands out as one of the most effective

This work was done during Yixin Liu's internship at Dolby Laboratories Inc. [1]Department of Computer Science, Lehigh University, Bethlehem, PA, USA [2]Dolby Laboratories Inc., San Francisco, CA, USA. Correspondence to: Lichao Sun <lis221@lehigh.edu>, Andrea Fanelli <andrea.fanelli@dolby.com>.

*Proceedings of the 42nd International Conference on Machine Learning*, Vancouver, Canada. PMLR 267, 2025. Copyright 2025 by the author(s).

proactive approaches. It involves embedding imperceptible perturbations into the audio that are both identifiable and traceable. Watermarking enables two key processes: detection, which verifies the presence of the watermark in an audio file, and attribution, which involves decoding a message that uniquely identifies the original creator.

Initialized by WavMark (Chen et al., 2023) and the seminal work of AudioSeal (San Roman et al., 2024), using end-to-end deep neural networks for learning to watermark audio content has demonstrated stronger robustness with minor quality degradation compared to the state-of-the-art hand-crafted watermarking method (Westerfeld, 2020). This is evident under challenging edits, such as EnCodec (Défossez et al., 2023). WavMark proposes an invertible neural network architecture with a 16-bit synchronization code and a 16-bit message code to jointly conduct detection and attribution. However, the brute-force decoding approach in WavMark is inefficient, and the invertible architecture limits the watermarking capacity under more challenging transformations (Chen et al., 2023). AudioSeal (San Roman et al., 2024) addresses these issues by decoupling the generator and detector and adding separate heads for detection and message decoding, which improves robustness to advanced transformations. However, this design boosts detection at the cost of lower attribution accuracy. In summary, *whether neural-network-based watermarking can achieve both robust detection and attribution is still an open problem.*

This paper identifies two key factors to bridge this gap: first, motivated by the shared-parameter architecture advantage of WavMark for boosting learning efficiency, and the disjoint generator-detector architecture of AudioSeal for robust capability, we introduce a blended architecture of partial parameter sharing between generator and detector, which jointly achieves both efficient learning and robustness. Specifically, we propose a cross-attention module that leverages a shared embedding table to facilitate message decoding in the detector part. Second, we design a simple yet effective conditioning mechanism that distributes the message temporally before injection, which further improves the learning efficiency. With these two key components, we observe significant gains in both detection robustness and attribution accuracy. To further improve watermark imperceptibility, we introduce a new per-tile TF masking loss. Specifically, we first compute masking energy with an asymmetric 2D kernel, identify the masked regions, and then use the masking energy as a weighting factor for computing a TF-weighted $\ell_2$ loss in the mel-spectrogram domain. With these efforts, we demonstrate state-of-the-art robustness across a wide range of audio editing transformations, while preserving superior perceptual quality (see Figure 1). Furthermore, under the more challenging task of generative model editing, we demonstrate that our approach is the only watermarking approach among the evaluated methods that

can conduct watermark detection even when edits of strong strength are applied. We summarize our main contributions as follows:

- Combining architectural advantages from prior works, we design partial parameter sharing between the neural generator and the detector, with an embedding table as the bridge and a cross-attention module in the detector as the core, to allow for more efficient learning and accurate message retrieval. Furthermore, we introduce a simple yet effective message conditioning module that distributes the latent message temporally, boosting the attribution learning efficiency.

- To improve perceptual quality, we introduce a new psychoacoustic-inspired time-frequency masking loss that captures per-tile masking effects. We compute masking energy with an asymmetric 2D kernel, identify the masked TF tiles, and assign lower loss weights to those tiles with a TF-weighted $\ell_2$ loss, achieving more imperceptible watermarking.

- We empirically show that our approach can achieve state-of-the-art performance in both detection and attribution with comparable perceptual quality and superior robustness. Furthermore, testing in a zero-shot manner on unseen generative editing transformations, our approach is the only one among the evaluated methods that maintains non-trivial detection under strong generative edits.

## 2. Related Work

**Audio Watermarking.** Audio watermarking has evolved significantly from traditional signal processing to modern deep learning approaches. Early rule-based methods focused on embedding watermarks in time or frequency domains through hand-crafted techniques (Zhang, 2020; Hu et al., 2020; Zhang et al., 2019; Qin et al., 2024). A notable example is AudiowMark (Westerfeld, 2020), which embeds a 128-bit message using convolutional coding and selective frequency band modifications. Although carefully engineered, hand-crafted methods often degrade under neural codec compression (Défossez et al., 2023). Deep neural networks (DNNs) have enabled more robust end-to-end watermarking systems that can generalize to unseen transformations (San Roman et al., 2024; Chen et al., 2023; Liu et al., 2023a). WavMark (Chen et al., 2023) introduced an invertible neural architecture for joint detection and attribution with 16-bit synchronization codes. While achieving strong performance, its brute-force decoding and architectural constraints limit scalability. AudioSeal (San Roman et al., 2024) addressed these limitations with a generator-detector design with separate detection and message decoding. The decoupled design improves detection but reduces

attribution accuracy. Our work focuses on enabling both robust detection and accurate attribution through a more efficient architecture design with a psychoacoustic-inspired quality loss.

**Source Attribution.** A central objective in copyright protection is the ability to trace and verify the origin of creative works, which remains challenging, especially for generative audio systems. Recent efforts have highlighted the necessity of robust source attribution mechanisms that work reliably across different transformations. For instance, Agnew et al. (Agnew et al., 2024) performed an extensive audit of popular audio datasets and revealed serious intellectual property infringements, underscoring the urgency for transparent dataset documentation and reliable authorship checks. In the music domain specifically, Barnett et al. (Barnett et al., 2024) advanced source attribution by leveraging audio embeddings to identify influential training data in generative music models, enabling a more transparent "musical roots" analysis. Such embedding-based similarity checks align with the broader push for dataset auditing, as reflected in Du et al. (Du et al., 2024), who argue for holistic copyright auditing mechanisms throughout the existing machine learning processes. In this work, we propose a neural watermarking system that advances in message decoding performance, marking an essential step toward robust source attribution.

## 3. Preliminaries

**Audio Watermarking.** Audio watermarking systems typically comprise two primary components: a generator $\mathcal{G}$ to embed watermark, and a detector $\mathcal{D}$ for recovering it. Let $\boldsymbol{x} \in \mathbb{R}^T$ be an audio signal of length $T$, and let $\boldsymbol{w} \in \{0,1\}^K$ be a binary watermark sequence. The generator $\mathcal{G} : \mathbb{R}^T \times \{0,1\}^K \to \mathbb{R}^T$ outputs a watermarked signal $\boldsymbol{x}_w = \mathcal{G}(\boldsymbol{x}, \boldsymbol{w})$, which should ideally preserve the audio's perceptual quality. The detector $\mathcal{D}$ is then responsible for two tasks. First, its detection head $\mathcal{D}^{\det} : \mathbb{R}^T \to [0,1]$ produces a probability indicating whether an input contains a valid watermark, which yields a final decision of presence or absence with a threshold. Second, its message decoding head $\mathcal{D}^{\mathrm{msg}} : \mathbb{R}^T \to [0,1]^K$ returns a vector indicating the probability of each bit being 1, which can be thresholded to reconstruct the embedded message. In practical settings, the watermarked audio may undergo various transformations $\mathcal{T}(\cdot)$ such as compression and cropping, resulting in a distorted signal $\boldsymbol{x}_w^T = \mathcal{T}(\boldsymbol{x}_w)$. Across different transformations, an ideal detector should be able to robustly detect and decode watermarks. In essence, the detection head should output probabilities close to 1 for watermarked signals (and close to 0 otherwise), while the message decoding head should recover a bit sequence $\tilde{\boldsymbol{w}}$ matching the original watermark message $\boldsymbol{w}$. In this paper, we focus on the per-sample level detection and attribution, which is a practi-

cal and widely adopted setting in real-world watermarking systems (Liu et al., 2024a).

**Learning to Watermark with Two-Headed Detector.** We first present the general formulation of a neural watermarking system with detection and decoding separation. A neural watermarking system is parameterized by $\Theta = (\Theta_{\mathcal{G}}, \Theta_{\mathcal{D}})$, where $\Theta_{\mathcal{G}}$ denotes the generator parameters and $\Theta_{\mathcal{D}}$ denotes the detector parameters. During training, the detection head typically first produces a logit for each audio input, which is then passed through a sigmoid function to yield a probability $p(\boldsymbol{x}) = \mathcal{D}_{\Theta}^{\det}(\boldsymbol{x}) \in [0,1]$. For a watermarked signal $\boldsymbol{x}_w^T$ and an unwatermarked signal $\boldsymbol{x}^T$, the detection head's objective is to correctly classify both, often formulated by maximizing the following expected log-likelihood:

$$\max_{\Theta_{\mathcal{D}^{\det}}} \left( \mathbb{E}_{\boldsymbol{x}, \boldsymbol{w}, \mathcal{T}} \left[ \log \mathcal{D}^{\det}(\boldsymbol{x}_w^T) \right] + \mathbb{E}_{\boldsymbol{x}, \mathcal{T}} \left[ \log \left( 1 - \mathcal{D}^{\det}(\boldsymbol{x}^T) \right) \right] \right),$$

where $\boldsymbol{x}_w = \mathcal{G}(\boldsymbol{x}, \boldsymbol{w})$ and $\boldsymbol{x}_w^T = \mathcal{T}(\boldsymbol{x}_w)$. During inference, the threshold $\tau$ is usually tuned on the validation set to control a certain level of false positive rate.

Meanwhile, the message decoding head produces $K$ probabilities $\mathcal{D}^{\mathrm{msg}}(\boldsymbol{x}_w^T) = [\mathcal{D}^{\mathrm{msg}}(\boldsymbol{x}_w^T)_1, \ldots, \mathcal{D}^{\mathrm{msg}}(\boldsymbol{x}_w^T)_K]$, where each entry represents the probability that the corresponding bit is 1. Its training objective is to maximize the probability of correctly predicting each bit of $\boldsymbol{w}$, which can be expressed as:

$$\max_{\Theta_{\mathcal{D}^{\mathrm{msg}}}} \mathbb{E}_{\boldsymbol{x}, \boldsymbol{w_k}, \mathcal{T}} \left[ \boldsymbol{w}_k \log \mathcal{D}^{\mathrm{msg}}(\boldsymbol{x}_w^T)_k + (1 - \boldsymbol{w}_k) \log(1 - \mathcal{D}^{\mathrm{msg}}(\boldsymbol{x}_w^T)_k) \right]. \tag{1}$$

By thresholding each $\mathcal{D}^{\mathrm{msg}}(\boldsymbol{x}_w^T)_k$ at 0.5, the decoded bit string $\hat{\boldsymbol{w}}$ is obtained, with the learning objective seeking $\hat{\boldsymbol{w}} = \boldsymbol{w}$ despite the distortion layer $\mathcal{T}$ applied to $\boldsymbol{x}_w$.

### 3.1. Overview of AudioSeal (San Roman et al., 2024)

To contextualize our architectural improvements, we first analyze AudioSeal's framework, which pioneered the disjoint generator-detector paradigm for neural watermarking. While it achieves strong detection robustness, its attribution limitations motivate key aspects of XATTNMARK.

**Disjoint Generator–Detector Architecture.** AudioSeal employs two separate networks for watermark generation and detection: given audio $\boldsymbol{x}$, the generator $\mathcal{G} = \{\mathcal{E}_{\mathrm{gen}}, \mathcal{J}_{\mathrm{gen}}\}$ is composed of an encoder $\mathcal{E}_{\mathrm{gen}}$ that encodes the audio into compact latent $\boldsymbol{h_x}$, and a decoder $\mathcal{J}_{\mathrm{gen}}$ that decodes the latent into the watermarked perturbation. The audio latent is obtained via $\boldsymbol{h_x} = \mathcal{E}_{\mathrm{gen}}(\boldsymbol{x}) \in \mathbb{R}^{t' \times H}$, where $t' = \lfloor T/\alpha \rfloor$, $\alpha$ is the temporal downsampling factor, and $H$ is the latent dimension. The secret message $w$ is injected into the latent space with a message encoder $\mathcal{M}$ that maps $\boldsymbol{w}$ to its latent $\boldsymbol{h}_w = \mathcal{M}(\boldsymbol{w})$. The modulated latent waveform is obtained by adding the message latent to the audio latent $\boldsymbol{h}(\boldsymbol{x}, \boldsymbol{w}) = \boldsymbol{h}_x + \boldsymbol{h}_w$, followed by the decoder

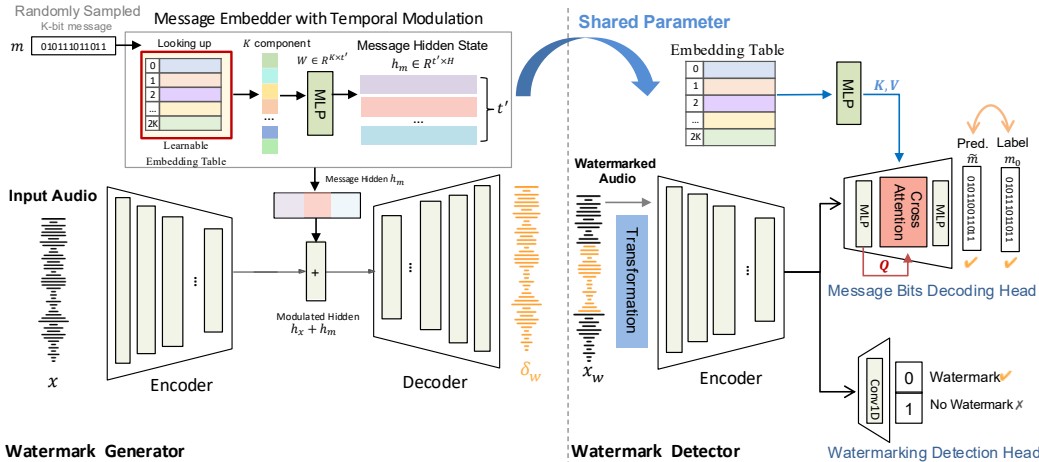

*Figure 2.* **System Diagram for XATTNMARK.** XATTNMARK consists of a watermark generator and a watermark detector, with a shared embedding table that facilitates message decoding through a cross-attention module. In the generator part, we first employ an encoder network to encode the audio latent and then apply a temporal modulation to hide the message. The modulated latent is then fed into a decoder to produce the watermark residual. In the detector part, a linear detection head is used for detecting the presence of watermarks, and a cross-attention module with a shared embedding table is used for message decoding.

network $\mathcal{J}_{\text{gen}}$ to produce the predicted watermarked perturbation $\boldsymbol{\delta_w} = \mathcal{J}_{\text{gen}}(\boldsymbol{h}(\boldsymbol{x}, \boldsymbol{w})) \in \mathbb{R}^{T \times 1}$, which is later applied on the original audio to produce the watermarked audio $\boldsymbol{x}_w = \boldsymbol{x} + \boldsymbol{\delta_w}$. Note that the decoder model $\mathcal{J}_{\text{gen}}$ shares a symmetric structure with the encoder $\mathcal{E}_{\text{gen}}$, with layers of residual transposed convolution for temporal up-sampling. The watermark detector $\mathcal{D} = \{\mathcal{E}_{\text{det}}, \mathcal{O}^{\text{dec}}\}$ is composed of an encoder $\mathcal{E}_{\text{det}}$ (sharing a similar structure with $\mathcal{E}_{\text{gen}}$ but with additional padding to match the original length $T$) and a unified decoding head $\mathcal{O}^{\text{dec}} \in \mathbb{R}^{H \times (1+K)}$ for both detection and message decoding. Given potential watermarked audio $\boldsymbol{x}_w^T$, the encoder $\mathcal{E}_{\text{det}}$ first processes the audio to obtain the latent $\tilde{\boldsymbol{h}}_x = \mathcal{E}_{\text{det}}(\boldsymbol{x}_w^T) \in \mathbb{R}^{T \times H}$. Then the decoding head $\mathcal{O}^{\text{dec}}$ is applied on the latent to obtain the logits for both detection and message decoding, i.e., $\mathcal{O}^{\text{dec}}(\tilde{\boldsymbol{h}}_x) \in \mathbb{R}^{T \times (1+K)}$.

**Multi-bit Message Conditioning $\mathcal{M}$.** Similar to the common practice of using learned embeddings for conditional generation (Peebles & Xie, 2023), AudioSeal uses an embedding table of only $2K$ entries, $\boldsymbol{E} \in \mathbb{R}^{2K \times H}$, to encode the total $2^K$ possible message states, which significantly reduces the storage complexity. Specifically, each bit $w_j \in \{0, 1\}$ for $j \in \{0, \ldots, K-1\}$ maps to index $I_j = 2j + w_j$, allowing separate representations for 0/1 states in each bit position with distinct embedding vectors. Given the position sequence $\boldsymbol{p} = [I_1, \ldots, I_K]$, the retrieved $K$ embedding vector sequence is $\boldsymbol{V}(\boldsymbol{w}) = [\boldsymbol{E}_{I_1}, \ldots, \boldsymbol{E}_{I_K}] \in \mathbb{R}^{K \times H}$. Then an average pooling operation on $\boldsymbol{V}(\boldsymbol{w})$ is applied to obtain the final message latent $\boldsymbol{h}_w = \frac{1}{K} \sum_{j=1}^{K} \boldsymbol{E}_{I_j} \in \mathbb{R}^H$. The message latent is repeated along the temporal axis and then added to the waveform latent, leading $\boldsymbol{h}(\boldsymbol{x}, \boldsymbol{w}) = \boldsymbol{h}_x + \boldsymbol{h}_w$.

# 4. Methodology

Despite AudioSeal achieving robust detection, the model struggles to perform accurate message decoding even without distortion (See Table 7). In this work, following the architecture backbone of AudioSeal, we identify and resolve two architectural limitations: disjoint generator-detector and the information bottleneck caused by embedding mean-pooling. Specifically, we first propose a cross-attention generator-detector watermarking system with a shared embedding table and temporal modulation to improve learning efficiency in message decoding. Then, to further improve watermark quality, we propose a psychoacoustic-aligned TF masking $\ell_2$ loss. The framework is shown in Figure 2.

## 4.1. Cross-Attention Generator-Detector Watermarking System with Shared Embedding Table

We observe that the fully disjoint architecture of AudioSeal ($\Theta_{\mathcal{G}} \neq \Theta_{\mathcal{D}}$) often converges fast for the watermark detection learning but struggles to learn the message decoding part efficiently and accurately. On the other hand, the full parameter-sharing architecture of WavMark ($\Theta_{\mathcal{G}} = \Theta_{\mathcal{D}}$) can achieve superior efficiency in learning both detection and message decoding but lacks enough robustness capability against various distortions (See App. C.2). This motivates the design of a hybrid architecture with partial parameter sharing between the generator and the detector, which facilitates more effective message decoding while preserving robust capability.

Our key design is to share the message conditioning module $\mathcal{M}$ between the generator and the detector, which helps

bridge the information flow between how the message is composed and how it can be reconstructed in the detector part. Specifically, the learnable part of the embedding table $\boldsymbol{E}$ in $\mathcal{M}$, which serves as the fundamental vector set for composing the message latent in the generator part, is now utilized as a reference when decomposing the latent to retrieve the message bits in the detector. To achieve this, we propose leveraging a cross-attention block (Vaswani et al., 2017) to use the embedding table $\boldsymbol{E}$ as a (key-value) reference for message decoding, given the query audio latent $\tilde{\boldsymbol{h}}_x$. We describe the decoding mechanism in detail in the following section.

**Message Decoding with Cross-Attention.** Ideally, for the given embedding table $\boldsymbol{E}$ with $2K$ entries and a watermarked audio $\boldsymbol{x}_w = \boldsymbol{x} + \mathcal{G}(\boldsymbol{x}, \boldsymbol{w})$, we want to reconstruct the original $K$ embedding vectors $[\boldsymbol{E}_{I_1}, \ldots, \boldsymbol{E}_{I_K}]$, that were used to compose the ground-truth message latent $\boldsymbol{h}_w$ and then feed them as context for the message decoding. To achieve this, we use an attention mechanism that transforms the embedding table into key and value using two different linear projections, and does attention-based vector merging with a query from the reconstructed latent representation $\tilde{\boldsymbol{h}}_x$. Specifically, since the nearby two entries in $\boldsymbol{E}$ represent one position with different bit states, merging them into one $2H$-dim vector represents the latent of each position. We transform this reshaped embedding matrix $\boldsymbol{E}' \in \mathbb{R}^{K \times 2H}$ into $\boldsymbol{K}, \boldsymbol{V}$ with two linear projections $\boldsymbol{W}_K, \boldsymbol{W}_V \in \mathbb{R}^{2H \times H}$, that is $\boldsymbol{K} = \boldsymbol{E}'\boldsymbol{W}_K$ and $\boldsymbol{V} = \boldsymbol{E}'\boldsymbol{W}_V$. Then, we demodulate the reconstructed waveform latent along the temporal axis $\tilde{\boldsymbol{h}}_x \in \mathbb{R}^{T \times H}$, to obtain the first version of raw prediction $\tilde{\boldsymbol{V}}_x$ for the original $K$ components. To do this, we first use a linear projection $\boldsymbol{W}_{dem} \in \mathbb{R}^{T \times K}$ for obtaining the query sequence $\tilde{\boldsymbol{h}}_x^{dem} = \boldsymbol{W}_{dem}^T \tilde{\boldsymbol{h}}_x \in \mathbb{R}^{K \times H}$ and follow with a linear query projection $\boldsymbol{W}_Q \in \mathbb{R}^{H \times H}$. Then we can utilize the embedding table $\boldsymbol{E}$ to further refine the final component prediction $\tilde{\boldsymbol{V}}_x$ with the following cross-attention mechanism:

$$
\begin{aligned}
\boldsymbol{Q} &= \tilde{\boldsymbol{h}}_x^{dem} \boldsymbol{W}_Q \in \mathbb{R}^{K \times H}, \\
\boldsymbol{K} &= \boldsymbol{E}' \boldsymbol{W}_K \in \mathbb{R}^{K \times H}, \\
\boldsymbol{V} &= \boldsymbol{E}' \boldsymbol{W}_V \in \mathbb{R}^{K \times H}, \\
\boldsymbol{A} &= \text{softmax}\left(\frac{\boldsymbol{Q}\,\boldsymbol{K}^\top}{\sqrt{H}}\right) \in \mathbb{R}^{K \times K}, \\
\tilde{\boldsymbol{V}}_x &= \text{act}\left(\boldsymbol{A}\,\boldsymbol{V}\right) \in \mathbb{R}^{K \times H},
\end{aligned}
\tag{2}
$$

where $H$ is the latent dimension, $K$ is the number of message bits, $t$ is the temporal dimension, and $\text{act}(\cdot)$ can be any activation function. In this study, we use exponential linear units (ELU) (Clevert et al., 2016). With the predicted $\tilde{\boldsymbol{V}}_x$, we further employ a linear projection layer $\boldsymbol{W}_{dec} \in \mathbb{R}^{H \times 1}$ with sigmoid activation function for the construction of the message decoding head. The final logit for message prediction bit is then obtained as $\hat{\boldsymbol{w}}_x = \sigma(\tilde{\boldsymbol{V}}_x \boldsymbol{W}_{dec})$, where $\sigma(\cdot)$

is the sigmoid activation function.

**Message Embedding via Temporal Modulation.** In AudioSeal, the message latent is obtained by a mean-pooling operation with temporal-axis repetition, $\boldsymbol{h}_w = \texttt{Repeat}\left(\frac{1}{K}\sum_{j=1}^{K}\boldsymbol{E}_{I_j}, t'\right) \in \mathbb{R}^{t' \times H}$. This approach can be viewed as injecting the message information mostly into the frequency domain, which greatly limits the message hiding capabilities. To improve upon this, knowing the input audio length $T$, we introduce a temporal message conditioning mechanism that employs a linear modulation layer $\boldsymbol{W}_M \in \mathbb{R}^{K \times t'}$ to obtain the message latent:

$$
\boldsymbol{h}_w = \boldsymbol{W}_M^\top \boldsymbol{V}(\boldsymbol{w}) \in \mathbb{R}^{t' \times H}.
\tag{3}
$$

This design avoids relying only on the frequency domain for distributing the $K$-bit message, but also significantly facilitates the learning process for message decoding (see Figure 6).

## 4.2. Psychoacoustic-Aligned Time-Frequency Masking

Achieving imperceptibility is a key requirement of any watermarking system. Ideally, a watermark should exploit the perception characteristics of the human auditory system so that the artifacts remain imperceptible. Among all the psychoacoustic effects, auditory masking (Gelfand, 2017; Holdsworth et al., 1988) offers strong theoretical guidance for designing perceptual losses in watermarking (Deng et al., 2023; Qin et al., 2019). While prior work like AudioSeal (San Roman et al., 2024) introduced perceptual losses such as TF-Loudness, these often adopt a coarse approach based on loudness differences within fixed TF tiles. Such methods face two main limitations: (i) they may not fully capture sophisticated auditory masking effects, particularly the interactions between masker and maskee signals across different tiles, and (ii) loudness differences alone can provide weak supervisory signals for guiding the watermark embedding, resulting in a coarse penalty distribution (visualized in App. C.9 in Figure 12). These limitations motivate our development of a more psychoacoustic-aligned TF-masking loss.

**Per-Tile Penalty with Asymmetric Temporal–Frequency Decay.** Let $\mathcal{S}_O(m, t)$ denote the magnitude of the original audio's mel-spectrogram, and the watermarked mel-spectrogram is $\mathcal{S}_W(m, t)$. To identify those strong "masker" tiles $\mathcal{M}$, we apply a magnitude threshold $\alpha_S$ to different timestamps on each frequency band ($\alpha_S = 0.8$):

$$
\mathcal{M}_{\text{masker}} = \{(m, t) \mid \mathcal{S}_O(m, t) > \alpha_S \max_{t'} \mathcal{S}_O(m, t')\}.
$$

Each index pair $(m_c, t_c) \in \mathcal{M}_{\text{masker}}$ acts as a masker, and we model its masking influence over neighboring tiles with a linear energy decay in the decibel domain. Specifically,

given mel-scale frequency radii $r_l^m$, $r_h^m$, and time-axis radius $r_b^t$, $r_f^t$, the local region of a masker is defined as:

$$R(m_c, t_c) = \{(m,t) \mid -r_l^m \leq \Delta m \leq r_h^m, -r_b^t \leq \Delta t \leq r_f^t\},$$

where $\Delta m = m - m_c$, and $\Delta t = t - t_c$. Aligned with the empirical studies (Necciari et al., 2016) that the post-masking region (forward masking) is usually longer than the pre-masking region (backward masking) in the temporal axis, we set asymmetric radii $r_b^t$ and $r_f^t$, with a forward masking region of 200ms and a backward masking region of 20ms. For the frequency-bin radius, we first compute the empirical critical bandwidth for each mel-bin using the formula from Zwicker & Fastl (2013), $W(m_c) = 25 + 75 \left(1 + 1.4 \left(\frac{\mathcal{F}(m_c)}{1000}\right)^2\right)^{0.69}$, where $\mathcal{F}(m_c)$ is the approximated center frequency obtained by converting each Mel bin value $m_c$ back to Hz using $\mathcal{F}(m_c) = 700 \cdot (10^{\frac{m_c}{2595}} - 1)$. Then the radii for each mel-bin are set as $r_l^m = r_h^m = r_b^m \cdot \gamma$, where $\gamma = W(m_c)/|\mathcal{F}(m_1) - \mathcal{F}(m_0)|$ is an adaptive scaling term, $V[m_i]$ denote the $i$-th mel-bin value, $r_b^m$ is a base radius that is set to be 3. With the dynamic radius scaled by the critical bandwidth – where higher-frequency components have wider masking ranges (Holdsworth et al., 1988) – we observe a better performance than using constant $\gamma$.

The physical structure of the cochlea determines that higher frequencies (located at the base of the cochlea) cause broader and stronger excitation patterns compared to the lower frequencies (located at the apex). To inject this bias of slower decay in the upward spread of masking (Zwicker & Fastl, 2013), we set the decaying term for the upward direction to be smaller than the one in the downward direction, i.e., $\alpha_f^+ \leq \alpha_f^-$. For the temporal decay, following Necciari et al. (2016), since the forward decay is slower, we set the forward decaying slope less steep than the backward one, i.e., $\beta_t^+ \leq \beta_t^-$. Given a masker $(m_c, t_c)$, the threshold energy $E_{\text{mask}}$ for the maskee tile $(m, t)$ is computed as:

$$E_{\text{mask}}(m, t; (m_c, t_c)) = 20 \log_{10} \mathcal{S}_O(m_c, t_c) - |\Delta E|,$$
$$\text{where} \quad \Delta E = \alpha_f^+ \max(0, \Delta m) + \alpha_f^- \min(0, \Delta m)$$
$$+ \beta_t^+ \max(0, \Delta t) + \beta_t^- \min(0, \Delta t), \quad (4)$$

In psychoacoustic modeling, particularly in audio compression (Bosi et al., 1997), the global masking threshold is determined by considering only the most dominant masker at each frequency component. Following this, the final masking threshold $E_{\text{mask}}^*(m, t)$ is computed as:

$$E_{\text{mask}}^*(m, t) = \max_{\substack{(m_c, t_c) \in \mathcal{M}_{\text{masker}}, \\ (m,t) \in R(m_c, t_c)}} E_{\text{mask}}(m, t; (m_c, t_c)).$$

This ensures that each maskee tile receives masking energy from its most perceptually dominant masker, while weaker maskers have negligible impact (Gelfand, 2017).

**Per-Tile-Weighted $\ell_2$ Loss Computation.** Psychoacoustic masking effectively applies only when the masker's masking energy surpasses the tile's energy. We therefore apply the masking threshold on watermarked mel-spectrogram at $20 \log_{10} \mathcal{S}_W(m, t)$, filtering the set of masked tiles as:

$$\mathcal{M}_{\text{maskee}} = \{(m,t) \mid E_{\text{mask}}^*(m, t) > 20 \log_{10} \mathcal{S}_W(m, t)\}.$$

To encourage the model to embed the watermark in those maskee regions, we design per-tile penalty terms which weight the $\ell_2$ difference (Chen et al., 2023) between the watermarked and original audio in mel-spectrogram space:

$$\lambda(m, t) = 1 + \mathbf{1}_{(m,t) \in \mathcal{M}_{\text{maskee}}} \cdot 10^{E_{\text{mask}}^*(m,t)/20}, \quad (5)$$
$$\mathcal{L}_{\text{TF}} = \sum_{(m,t)} \frac{\|\mathcal{S}_W(m, t) - \mathcal{S}_O(m, t)\|_2^2}{\lambda(m, t)}, \quad (6)$$

where $10^{E_{\text{mask}}^*(m,t)/20}$ is the masking energy, and the weighting term $\lambda(m, t)$ is larger for tiles in the masking region $\mathcal{M}_{\text{maskee}}$, effectively allowing more watermark signal to be embedded in those locations. For non-masked tiles, $\lambda(m, t) = 1$ enforces standard $\ell_2$ loss.

**Other Auxiliary Perceptual Loss.** Following San Roman et al. (2024), we use a $\ell_1$ constraint on the watermark signal $\delta_w$ to ensure waveform-domain smoothness, and we incorporate a multi-scale Mel spectrogram loss $\mathcal{L}_{\text{msspec}}$ (e.g., as in Défossez et al. (2023)) to manage frequency-domain fidelity. Furthermore, we also adopt adversarial losses $\mathcal{L}_{adv}$ on multi-scale STFT spectrograms for perceptual improvement, but with a lower weight $\lambda_{\text{adv}} = 1$ (while AudioSeal uses $\lambda_{\text{adv}} = 4$). The reason behind this is that aggressively pushing the watermarked audio distribution to be close to the clean audio distribution in an adversarial sense may penalize desirable watermark characteristics (e.g., certain musical "remixes" can produce satisfactory imperceptible watermarks, but still yield high adversarial loss). We observe that lowering the adversarial weight allows the model to focus more on the psychoacoustic aspects to develop a human-centric, imperceptible watermark.

## 5. Experiments

### 5.1. Experimental Setup

Following prior works (San Roman et al., 2024; Chen et al., 2023), we use a sampling rate of 16 kHz and one-second mono samples for training ($T = 16000$) under 16 diverse audio editing transformations. We train the models on a mixed audio dataset of 4100 hours containing speech (3016 hours VoxPopuli (Wang et al., 2021) and 100 hours LibriSpeech (Panayotov et al., 2015)), music (9 hours Music-Caps (Agostinelli et al., 2023) and 880 hours Free Music Archive (Defferrard et al., 2016)), and sound effects (98 hours AudioSet (Gemmeke et al., 2017)). For evaluation,

*Table 1.* The accuracy of detection and attribution across different editing operations. For detection, we also report the True Positive Rate (TPR) and False Positive Rate (FPR) where the threshold is selected by Youden's Index (Youden, 1950) on a balanced validation set.

| Edit | AudiowMark Det. (TPR/FPR) | Att. | WavMark Det. (TPR/FPR) | Att. | TimbreWM Det. (TPR/FPR) | Att. | AudioSeal Det. (TPR/FPR) | Att. | XATTNMARK Det. (TPR/FPR) | Att. |
|---|---|---|---|---|---|---|---|---|---|---|
| Identity | 1.00 (1.00/0.00) | 1.00 | 1.00 (1.00/0.00) | 1.00 | 0.995 (0.99/0.00) | 0.93 | 1.00 (0.99/0.00) | 0.69 | 0.995 (0.99/0.00) | 1.00 |
| Bandpass | 1.00 (1.00/0.00) | 1.00 | 1.00 (1.00/0.00) | 1.00 | 0.985 (0.97/0.00) | 0.93 | 1.00 (0.99/0.00) | 0.31 | 0.995 (0.99/0.00) | 0.99 |
| Boost | 1.00 (1.00/0.00) | 1.00 | 1.00 (1.00/0.00) | 1.00 | 0.98 (0.96/0.00) | 0.91 | 1.00 (0.99/0.00) | 0.50 | 0.995 (0.99/0.00) | 1.00 |
| Duck | 1.00 (1.00/0.00) | 1.00 | 0.995 (0.99/0.00) | 1.00 | 0.97 (0.94/0.00) | 0.89 | 1.00 (0.99/0.00) | 0.56 | 0.995 (0.99/0.00) | 1.00 |
| Echo | 1.00 (1.00/0.00) | 1.00 | 1.00 (1.00/0.00) | 1.00 | 0.945 (0.89/0.00) | 0.91 | 1.00 (0.99/0.00) | 0.38 | 0.995 (0.99/0.00) | 0.99 |
| Highpass | 1.00 (1.00/0.00) | 1.00 | 0.95 (0.90/0.00) | 1.00 | 0.985 (0.97/0.00) | 0.93 | 1.00 (0.99/0.00) | 0.31 | 0.995 (0.99/0.00) | 1.00 |
| Lowpass | 1.00 (1.00/0.00) | 1.00 | 1.00 (1.00/0.00) | 1.00 | 0.985 (0.97/0.00) | 0.90 | 1.00 (0.99/0.00) | 0.56 | 0.995 (0.99/0.00) | 1.00 |
| MP3 | 0.94 (0.88/0.00) | 1.00 | 0.805 (0.61/0.00) | 1.00 | 0.95 (0.91/0.01) | 0.86 | 1.00 (0.99/0.00) | 0.38 | 0.995 (0.99/0.00) | 1.00 |
| Pink Noise | 1.00 (1.00/0.00) | 1.00 | 1.00 (1.00/0.00) | 1.00 | 0.995 (0.99/0.00) | 0.92 | 1.00 (0.99/0.00) | 0.75 | 0.995 (0.99/0.00) | 1.00 |
| White Noise | 1.00 (1.00/0.00) | 1.00 | 1.00 (1.00/0.00) | 1.00 | 0.975 (0.95/0.00) | 0.90 | 1.00 (0.99/0.00) | 0.56 | 0.995 (0.99/0.00) | 1.00 |
| Smooth | 1.00 (1.00/0.00) | 1.00 | 0.96 (0.92/0.00) | 0.88 | 0.98 (0.96/0.00) | 0.80 | 1.00 (0.99/0.00) | 0.19 | 0.995 (0.99/0.00) | 1.00 |
| Speed | 0.50 (0.00/0.00) | 0.00 | 0.50 (0.00/0.00) | 0.00 | 0.515 (0.07/0.04) | 0.18 | 0.61 (0.36/0.15) | 0.00 | 0.995 (0.99/0.00) | 0.03 |
| Resample | 1.00 (1.00/0.00) | 1.00 | 1.00 (1.00/0.00) | 1.00 | 0.98 (0.96/0.00) | 0.92 | 1.00 (0.99/0.00) | 0.56 | 0.995 (0.99/0.00) | 1.00 |
| AAC | 1.00 (1.00/0.00) | 1.00 | 1.00 (1.00/0.00) | 1.00 | 0.975 (0.95/0.00) | 0.90 | 1.00 (0.99/0.00) | 0.12 | 0.995 (0.99/0.00) | 0.88 |
| EnCodec (nq=16) | 0.50 (0.00/0.00) | 0.00 | 0.805 (0.61/0.00) | 0.00 | 0.625 (0.50/0.25) | 0.07 | 1.00 (0.99/0.00) | 0.31 | 0.965 (0.93/0.00) | 0.99 |
| Crop | 0.965 (0.93/0.00) | 1.00 | 0.995 (0.99/0.00) | 1.00 | 0.96 (0.92/0.00) | 0.85 | 1.00 (0.99/0.00) | 0.12 | 0.975 (0.98/0.03) | 1.00 |
| **Average** | 0.929 (0.859/0.000) | 0.88 | 0.918 (0.836/0.000) | 0.87 | 0.925 (0.869/0.019) | 0.80 | 0.971 (0.950/0.010) | 0.39 | **0.9919** (0.9856/0.0019) | **0.93** |

we use a held-out test set from MusicCaps of size 100. For each audio file, we embed 100 distinct messages, resulting in 10k unique watermarked audio samples. Each of these samples is then subjected to 16 different audio transformations, leading to a total of 160k evaluated instances. The audio duration is set as 5s by default in evaluation. The loss weights are set as: $\lambda_{TF} = 1$, $\lambda_{adv} = 1$, $\lambda_{\ell_1} = 0.1$, $\lambda_{msspec} = 2$, $\lambda_{detect} = \lambda_{message} = 10$. We use the Adam optimizer (Kingma & Ba, 2015) with learning rate 1e-5, $\beta_1 = 0.4$, $\beta_2 = 0.9$, and Exponential Moving Average (EMA) (Tarvainen & Valpola, 2017) with decay factor of 0.99 updated at every step. We train for 73k steps with batch size 16 and latent size $H = 32$. To boost the sampling of the transformation efficiency, we update the sampling probability of each transformation every 1000 steps on the validation set, adjusting it based on the validation accuracy of each transformation.

For the attribution experiment, we follow the simulation protocol in San Roman et al. (2024) that defines a message pool of size $N$ ($\in \{100, 1000, 10000\}$), where each message is uniquely associated with a different user. As $N$ increases, the message length (in bits) also grows, making attribution more challenging due to the increased complexity of distinguishing individual messages. During decoding, the message is retrieved from the pool by selecting the one with the closest Hamming distance. We define attribution accuracy as *the fraction of correctly attributed messages among all audio inputs that were positively detected as watermarked*. We compare to four state-of-the-art baselines: AudiowMark (Westerfeld, 2020), WavMark (Chen et al., 2023), TimbreWM (Liu et al., 2023a), AudioSeal (San Roman et al., 2024). Please refer to App. A for more details.

## 5.2. Detection and Attribution Effectiveness

**Robustness to Standard Edits.** We present the detection and attribution performance across a comprehensive suite of standard audio editing operations in Table 1. For detection, we report both the overall accuracy and the true/false positive rates (TPR/FPR). For attribution, we report the averaged performance across different user numbers (see Figure 5 for the decomposed result). Our method achieves new state-of-the-art performance on both tasks, maintaining high detection accuracy (99.19% average) and attribution accuracy (93%) on average across transformations. Notably, while AudioSeal achieves strong detection performance (97.1% average), it struggles to perform effective attribution (39% average). The traditional approach, AudiowMark, exhibits more balanced detection-attribution trade-offs (around 88% attribution) but lower overall detection performance (92.9% average). Under the speed-change transformation, our method achieves 99.5% detection accuracy while all other methods degrade to near random-guess levels (50-61%). While attribution remains challenging under speed change, we show that the attribution performance can be partially restored via a speed-reversion layer (see App. C.3.1 for details). Figure 5 further validates that XATTNMARK consistently maintains high attribution accuracy even with increasing message pool sizes. Furthermore, we conduct statistical tests to validate the statistical significance of the improvement in attribution across different transformations (App. C.14 and App. C.3 for more details). In summary, we demonstrate that XATTNMARK is more robust under standard edits for both detection and attribution.

**Robustness to Generative Edits.** Beyond the standard audio transformations that are seen during training, audio generative editing is one particularly challenging transfor-

*Table 2.* The detection performance (Accuracy (TPR/FPR)) under two generative edits applied with different editing strengths $t$.

| System | $t$ | AudiowMark | WavMark | TimbreWM | AudioSeal | XATTNMARK |
|---|---|---|---|---|---|---|
| **Stable Audio** | 10 | 0.50 (0.50/0.50) | 0.50 (0.50/0.50) | 0.47 (0.81/0.88) | 0.69 (0.63/0.25) | **0.94** (0.94/0.06) |
| | 70 | 0.50 (0.50/0.50) | 0.50 (0.50/0.50) | 0.53 (0.19/0.13) | 0.59 (0.75/0.56) | **0.91** (0.88/0.06) |
| | 110 | 0.50 (0.50/0.50) | 0.50 (0.50/0.50) | 0.53 (0.56/0.50) | 0.59 (0.94/0.75) | **0.91** (0.88/0.06) |
| **Audio LDM2** | 10 | 0.50 (0.50/0.50) | 0.50 (0.50/0.50) | 0.53 (0.81/0.75) | 0.59 (0.94/0.75) | **0.94** (0.94/0.06) |
| | 70 | 0.50 (0.50/0.50) | 0.50 (0.50/0.50) | 0.50 (0.13/0.13) | 0.59 (0.94/0.75) | **0.94** (0.94/0.06) |
| | 110 | 0.50 (0.50/0.50) | 0.50 (0.50/0.50) | 0.50 (0.00/0.00) | 0.59 (0.94/0.75) | **0.94** (0.94/0.06) |

*Table 3.* Detection performance and audio quality metrics after HSJA-based (black-box) adversarial attacks on the waveform and spectrogram domains on AudioMarkBench dataset (Liu et al., 2024a).

| Diff. Setup | AudioSeal | | | | XATTNMARK | | | |
|---|---|---|---|---|---|---|---|---|
| | Acc. | PESQ | SISNR | ViSQOL | Acc. | PESQ | SISNR | ViSQOL |
| Waveform | 0.15 | 1.14 | 8.97 | 2.61 | 0.68 | 2.80 | 17.79 | 3.34 |
| Spectrogram | 0.15 | 1.05 | -17.82 | 2.45 | 0.36 | 1.56 | -24.08 | 2.37 |
| #Q=100 | 0.15 | 1.14 | 8.97 | 2.61 | 0.68 | 2.80 | 17.79 | 3.34 |
| #Q=200 | 0.15 | 1.13 | 8.95 | 2.64 | 0.57 | 2.30 | 6.93 | 2.91 |
| #Q=500 | 0.15 | 1.13 | 8.72 | 2.63 | 0.47 | 1.97 | 1.66 | 2.55 |

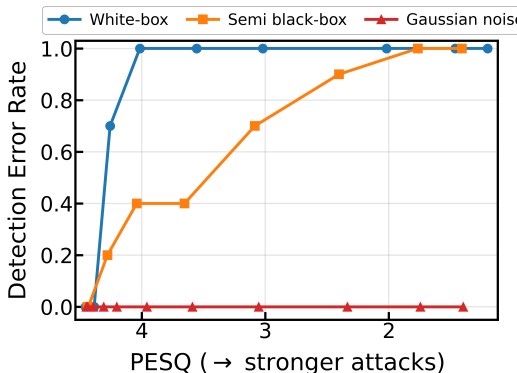

*Figure 3.* Watermark-removal attacks under three different knowledge settings. PESQ is measured between attacked audios and genuine ones (PESQ < 4 indicates strong quality degradation).

mation that watermarking systems might have to endure at deployment (Liu et al., 2024b). To simulate this, we use two state-of-the-art audio generative models, AudioLDM2 (Liu et al., 2024b) and Stable Audio (Evans et al., 2024), with a text-guided DDIM inversion method proposed in ZETA (Manor & Michaeli, 2024). We test various editing strengths $t \in \{10, 70, 110\}$, which represent the diffusion forward step when using DDIM inversion. As shown in Table 2, AudiowMark and WavMark degrade to a random-guess level performance, while TimbreWM and AudioSeal show inferior performance across different editing strengths, around 50-60%. In contrast, our method maintains consistently high detection accuracy, averaging 91-94% across all editing strengths, and is consistent across both generative models. While XAttnMark demonstrates strong detection robustness (e.g., >90% accuracy) against generative editing attacks in a zero-shot manner, achieving similarly robust attribution performance under such strong edits remains a challenge and an important direction for future work. Nevertheless, our detection performance marks a significant advancement. This demonstrates that XATTNMARK generalizes better to unseen generative edits compared to existing methods. Please refer to the App. C.4 for more details.

**Robustness to Adversarial Watermark Removal.** Recent work by Liu et al. (2024a) demonstrates that adversarial perturbations can be crafted to make watermarks undetectable, even with only black-box access to the detection model. To evaluate robustness against such attacks, we employ the black-box HopSkipJumpAttack (HSJA) (Chen et al., 2020), which iteratively crafts minimal perturbations in either the waveform or spectrogram domain to fool the detector. As shown in Table 3, we first evaluate HSJA attacks in both domains with a query budget of $Q = 100$. While the attacks successfully reduce detection accuracy, the attacks

also significantly degrade audio quality in the process. In the waveform domain, increasing the query budget from $Q = 100$ to $Q = 500$ further reduces detection accuracy from 0.68 to 0.47. However, across all experimental configurations, XATTNMARK consistently outperforms AudioSeal in terms of both detection robustness and perceptual quality preservation. Aligning with San Roman et al. (2024), we further evaluate XATTNMARK against white-box, semi-black-box (using a re-trained surrogate model with the same architecture but different initialization), and Gaussian noise attacks, with results shown in Figure 3. The results reveal that model vulnerability increases with attacker knowledge, highlighting the importance of keeping detector models proprietary to mitigate watermark removal threats.

**Extending to Watermark Localization.** We further extend XATTNMARK to perform watermark localization following the Brute Force Detection (BFD) method used in WavMark (Chen et al., 2023). This is achieved by employing a sliding window detection mechanism. Given that our model operates on 1-second audio segments and incorporates transformations robust to time shifts, we can distribute the per-segment detection probability to a finer, per-frame level by utilizing multiple overlapping detection windows. As shown in Figure 4, XATTNMARK achieves localization performance comparable to AudioSeal and significantly outperforms WavMark, in terms of both sample-level F1 score and Intersection Over Union (IoU) metrics. Furthermore, we also demonstrate the effectiveness in the case where the duration of the watermark segment is fixed at 1 second, and the total audio length varies from 2 seconds to 10 seconds. Please refer to the App. C.10 for more results.

### 5.3. Quality and Stealthiness Assessment

We evaluate the following objective quality metrics of the watermarked audio: Scale Invariant Signal to Noise

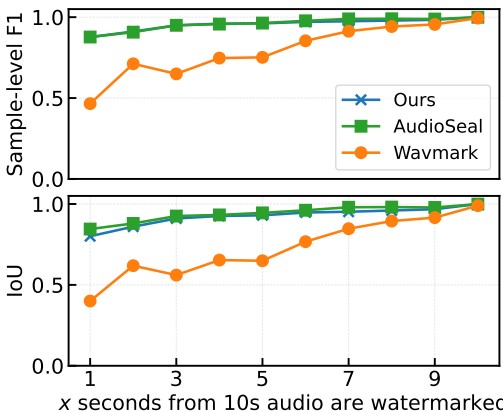

*Figure 4.* **Watermark localization results** across different durations of watermarked windows in terms of sample-level F1 score and Intersection Over Union (IoU) metrics (↑ is better).

*Table 4.* **Audio Quality Metrics**. Comparison of objective perceptual quality metrics across watermarking methods.

| Methods | SI-SNR ↑ | PESQ ↑ | STOI ↑ | ViSQOL ↑ | $L(\delta_w)$ ↓ |
|---|---|---|---|---|---|
| AudiowMark | 27.88 | **4.59** | 0.988 | **4.72** | -44.96 |
| WavMark | **36.35** | 4.43 | 0.985 | 4.62 | -53.01 |
| TimbreWM | 26.45 | 4.29 | 0.974 | 4.63 | -43.69 |
| AudioSeal | 25.32 | 4.51 | 0.990 | 4.72 | -44.51 |
| XATTNMARK | 29.00 | 4.43 | **1.000** | 4.56 | **-54.63** |
| w/o $\mathcal{L}_{TF}$ | 19.64 | 4.25 | 0.990 | 4.26 | -52.63 |
| w/o Adaptive $\gamma$ | 21.89 | 4.22 | 0.991 | 4.24 | -54.30 |

Ratio (SI-SNR), as well as PESQ (Rix et al., 2001), ViSQOL (Hines et al., 2012) and STOI (Taal et al., 2010). Furthermore, we also report the loudness of watermark residual $L(\delta_w)$ according to ITU-R BS.1770-4 (International Telecommunication Union, 2011), which is an important metric for watermarking stealthiness, especially under residual-based detection (Yang et al., 2024). As shown in Table 4, XATTNMARK achieves competitive performance across all perceptual quality metrics. While WavMark achieves the highest SI-SNR (36.35 dB), XATTNMARK excels in perceptual quality measures, achieving the best STOI score (1.000) and the lowest watermark residual loudness (-54.63 LUFS). For the PESQ and ViSQOL scores, XATTNMARK achieves competitive performance, with a PESQ score of 4.43 and a ViSQOL score of 4.56. We also conducted a subjective listening test using the MUSHRA protocol (International Telecommunication Union, 2014) with 12 participants, and the results suggest that XATTN-MARK achieves a comparable quality with AudioSeal scoring around 91 (while ground truth scores around 95, more details in the App. C.1 and App. C.9). In summary, XAT-TNMARK achieves superior utility while maintaining competitive perceptual quality.

## 5.4. Ablation Study

We first conduct ablation on the cross-attention module and temporal modulation layer to showcase their effectiveness for boosting learning in message decoding. As shown in Figure 6, our approach demonstrates superior learning efficiency, reaching approximately 98% accuracy. When removing the modulation component, performance drops significantly to around 60%. More dramatically, ablating the cross-attention mechanism causes accuracy to drop to random-guess levels (50%), highlighting its crucial role in the architecture. Our ablation studies highlight the contributions of key architectural components. Notably, the partial parameter sharing enabled by the cross-attention mechanism is particularly impactful for improving message decoding efficiency and accuracy. While the temporal conditioning module also contributes, the ability of the detector to directly attend to shared message embeddings via cross-attention appears to be a more dominant factor in achieving robust attribution compared to ablations without it. Furthermore, we ablate the proposed TF loss $\mathcal{L}_{TF}$ and the adaptive bandwidth with constant weight $\gamma = 1$. As shown in Table 4, both of them contribute to perceptual quality.

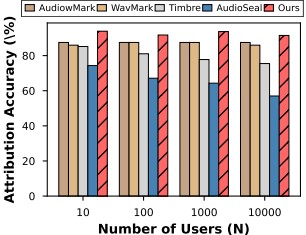 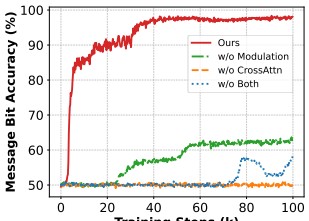

*Figure 5.* Attribution accuracy with different #Users.

*Figure 6.* Ablation study on the proposed architecture.

## 6. Conclusion

In this paper, we propose a neural audio watermarking framework that achieves both reliable watermark detection and accurate message attribution under challenging transformations, including generative editing. By integrating partial parameter sharing, a cross-attention-based detector, and a temporal conditioning module, our approach improves robustness and decoding accuracy compared to existing neural watermarking methods. Additionally, the psychoacoustic-aligned TF masking loss enables quality-preserving watermarking. To our knowledge, this is the first method to demonstrate non-trivial watermark detection under generative editing, marking a significant step toward reliable content attribution in generative AI settings. Future work includes developing stronger watermarking for robust attribution under speeding and various generative editing, and leveraging spatial psychoacoustics to improve the perceptual quality and support spatial audio watermarking.

## Acknowledgments

This work was supported in part by Dolby Laboratories Inc. and Lehigh University. In this work, Yixin Liu and Lichao Sun were partially supported by the National Science Foundation Grants CRII-2246067, ATD-2427915, NSF POSE-2346158, and Lehigh Grant FRGS00011497. This work is also supported by the Delta/DeltaAI systems at NCSA and the Bridges-2 system at PSC through allocation CIS240308 from the Advanced Cyberinfrastructure Coordination Ecosystem: Services & Support (ACCESS) program, supported by National Science Foundation grants 2138259, 2138286, 2138307, 2137603, and 2138296. We also thank Leslie Famularo from Dolby Laboratories Inc. for helpful discussions and the anonymous reviewers for their constructive feedback.

## Impact Statement

This research addresses pressing concerns regarding media authenticity and intellectual property protection stemming from the rise of generative AI. Our work introduces an improved audio watermarking system. The system's value lies in its ability to reliably detect embedded identifying marks (watermarks) and accurately trace their origin, even when audio has been substantially altered by sophisticated AI tools. This capability makes our technology a valuable asset in combating deepfake audio and the spread of misinformation. Furthermore, it offers enhanced protection for the rights of audio content creators.

Like many technologies, advanced watermarking holds the potential for dual use. Although designed to protect and authenticate, such systems could potentially be misused by malicious actors or institutions for purposes such as unauthorized tracking or censorship if implemented irresponsibly. Recognizing these risks, we strongly advocate for transparency and informed user consent in the deployment of such systems. The development of clear ethical guidelines and robust policies is crucial to prevent abuse.

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

# A. Implementation Details

**Model Architecture.** Following AudioSeal (San Roman et al., 2024), we leverage a pair of convolutional encoder-decoder models that operate on the waveform domain to generate and detect watermarks. The watermark generator consists of a waveform encoder and decoder, both utilizing components from EnCodec (Défossez et al., 2023). The encoder employs a 1D convolution containing 32 channels and a kernel size of 7, then uses four convolutional blocks. Each of these blocks contains a residual unit and a down-sampling layer, utilizing convolution with stride $S$ and kernel size $K = 2S$. The residual unit contains two kernel-3 convolutions with a skip-connection, with channels doubling during down-sampling. The encoder ends with a two-layer LSTM and a final 1D convolution having a kernel size of 7 and 128 channels. The stride $S$ values used are (2, 4, 5, 8), and residual units utilize the Exponential Linear Unit (ELU) as a nonlinear activation. The decoder reflects the encoder's structure but employs transposed convolutions, with strides in the opposite order.

The detector contains an encoder, transposed convolution, and linear layer for the detection head. The encoder utilizes the same structure as the one from the generator, but with different weights. The transposed convolution includes $H$ output channels and upscales the activation map to match the original audio dimension, yielding an activation map shaped $(T, H)$. This frame-level latent state serves both the detection head and the message decoding head. For the *detection head*, the linear layer serves to reduce the dimensionality of $H$ to two, followed by a softmax function that produces per-sample detection probability scores. To obtain the sample-level detection probability from frame-level logits, we apply temporal averaging (mean pooling) across the time dimension. Additional details on the *message decoding head* are provided in the next section.

*Cross-Attention Message Decoding Head.* The message decoding head aims to decode the embedded message bits by leveraging the shared embedding table from the generator. The key idea is to have a "message vocabulary" to look up when decoding the message bits. To this end, we employ a single-head cross-attention layer with the message latent state as a query and the units in the embedding table as key-value pairs. Note that the reconstructed waveform latent is first passed through a fully connected layer to decompose the time dimension and obtain a $K$-dimensional message latent state as a query. For the embedding table part, we reshape the embedding table to a $2H$-dimensional vector by combining the two vicinal units that correspond to the same message bit into one $2H$-dimensional vector. All the linear projections $Q, K, V$ are designed to map the latent dimension to $H$ dimensions. After the cross-attention operation, we apply an ELU activation function followed by a final linear projection $W \in \mathbb{R}^{H \times 1}$ to obtain the final per-bit message logits for the logistic regression.

**Training and Inference Details.** During the training stage, we employ a curriculum learning method to first disable the gradient-based quality balancing term, guiding the model to focus more on learning watermark decoding at the beginning of training. After 59000 training steps, we enable the quality balancing term to help the model improve the watermark auditory transparency with more dynamic scaling across different perceptual loss terms. To boost the sampling of the transformation efficiency, we update the sampling probability of each transformation every 1000 steps on the validation set, adjusting it based on the validation accuracy of each transformation. In detail, the sampling ratio $p_g$ for transformation $g$ is given by $p_g = \frac{1 - \text{acc}_g}{\sum_g 1 - \text{acc}_g} + \epsilon$, where $\text{acc}_g$ is the accuracy of the model on transformation $g$ and $\epsilon$ is a small constant for keeping the sampling ratio non-zero. We observe that it boosts the learning efficiency in attribution with at least 2x improvement under the main setup. The training checkpoint at 73000 is selected based on its balanced performance between quality and utility scores on the validation set. During the inference stage, for given audio with arbitrary duration, we first pad and split the input audio into multiple 1s audio segments and then apply the watermarking model on each chunk. Audio chunks are then concatenated to obtain the final watermarked audio.

**Perceptual Loss.** Following AudioSeal (San Roman et al., 2024), we employ a pool of perceptual loss terms to guide the model to learn the watermarking task. The perceptual loss terms include: i) multi-scale mel-spectrogram loss $\mathcal{L}_{\text{msspec}}$ that computes both L1 loss on linear-scale mel-spectrograms and MSE loss on log-scale mel-spectrograms across multiple FFT window sizes (from $2^6$ to $2^{11}$), with each scale weighted by $\sqrt{2^i - 1}$ for scale $i$; ii) feature matching loss $\mathcal{L}_{\text{feat}}$ that minimizes L1 distance between intermediate feature maps extracted from the adversarial discriminator, averaged across all layers. iii) waveform L1 loss $\mathcal{L}_{\text{waveform}}$ that minimizes L1 distance between the original waveform and the watermarked waveform; iv) our proposed psychoacoustic-aligned TF masking loss $\mathcal{L}_{\text{mask}}$, which captures the weighted $\ell_2$ loss in the mel-spectrograms domain, using a per-tile weight obtained through simulation of the masker energy decaying in the time-frequency domain.

For the adversarial discriminator, we adopt a multi-scale STFT discriminator architecture that operates on the complex STFT representations of the audio at different scales. Specifically, it consists of multiple sub-discriminators, each processing the STFT with different FFT sizes (512, 1024, 2048), hop lengths (128, 256, 512), and window sizes (512, 1024, 2048). Each sub-discriminator first computes the STFT, concatenates the real and imaginary components along the channel dimension,

and processes them through a series of 2D convolutions with increasing dilation rates (1,2,4). The feature maps from these convolutional layers are used for the feature matching loss. These losses jointly optimize for perceptual quality by matching both time-frequency characteristics and learned audio representations.

**Evaluation Setup.** For the attribution experiment, following San Roman et al. (2024), we set up a pool of $N$ potential users, where each user's sequential bit message is derived from their ordinal number, i.e., the $i$-th user's message is given by $w_i = \text{binary}(i)$. This creates a space of all potential messages $E_W$. The attribution is achieved by retrieving the message in $E_W$ with the smallest Hamming distance. Compared to bit-wise message accuracy, this metric provides a more practical measure of message retrieval performance, aligning better with real-world watermarking applications and attribution use cases. The message length (in bits) determines the maximum number of uniquely identifiable users. Specifically, for a message of $K$ bits, the system can support up to $2^K$ unique users. In our experiments, we use a 16-bit message, enabling attribution across $2^{16} = 65{,}536$ users at maximum.

### A.1. Robustness Augmentations

Following AudioSeal (San Roman et al., 2024), we apply these audio editing augmentations during training and evaluation:

- **Bandpass Filter:** Simulates frequency-selective audio equipment by allowing only mid-range frequencies. Allows frequencies between 300Hz-8000Hz to pass through.
- **Highpass Filter:** Removes bass frequencies to simulate poor bass response. Cuts frequencies below 500Hz.
- **Lowpass Filter:** Removes high frequencies to simulate muffled audio. Cuts frequencies above 5000Hz.
- **Speed:** Simulates playback speed variations. Changes speed by random factor 0.8-1.2.
- **Resample:** Robustness to sample rate conversion. Upsamples to 32kHz, then downsamples back to the original rate.
- **Boost Audio:** Simulates volume increase. Amplifies by factor 1.2.
- **Duck Audio:** Simulates volume decrease. Reduces volume by a factor of 0.8.
- **Echo:** Simulates room acoustics and reverberation. Adds delayed copy with 0.1-0.5s delay and 0.1-0.5 volume.
- **Pink Noise:** Adds realistic environmental noise. Adds pink noise with std 0.01.
- **White Noise:** Adds Gaussian noise with standard deviation 0.001.
- **Smooth:** Simulates low-quality audio processing. Moving average filter with window size 2-10.
- **AAC:** Robustness to common lossy compression. AAC encoding at 128kbps.
- **MP3:** Robustness to common lossy compression. MP3 encoding at 128kbps.
- **EnCodec:** Tests neural audio codec compression. Resamples to 24kHz, encodes with 16 streams ($nq = 16$), resamples to 16kHz.
- **Crop:** Robustness to audio truncation, padding, and in-batch audio mixing. While San Roman et al. (2024) uses this to obtain a localization mask, we implement crop as one of the edits to gain cropping-based robustness. Specifically, we first randomly select $k$ starting points, then modify $T/2k$ consecutive samples in one of four ways:
  - Revert to original audio (40% probability)
  - Replace with zeros (20% probability)
  - Substitute with different audio from the same batch (20% probability)
  - Leave unmodified (20% probability)

### A.2. Dataset Details

We utilize a mixed dataset of 4100 hours in total for training, which contains 100.59-hour LibriSpeech (Panayotov et al., 2015), 98.53-hour AudioSet (Gemmeke et al., 2017), 879.29-hour Free Music Archive (Defferrard et al., 2016), 9-hour MusicCaps (Agostinelli et al., 2023), and 3,016.43-hour VoxPopuli (Wang et al., 2021). Each dataset is described as follows:

**LibriSpeech.** LibriSpeech (Panayotov et al., 2015) is an English speech dataset derived from audiobooks in the LibriVox project. The dataset contains approximately 1000 hours of read English speech sampled at 16 kHz, with careful segmentation and alignment. The audio is paired with transcribed text, making it suitable for speech recognition tasks. We used a 100.59-hour subset of the full 1000-hour dataset for training.

**AudioSet.** AudioSet (Gemmeke et al., 2017) is a large-scale dataset containing 2,084,320 human-labeled 10-second sound clips drawn from YouTube videos. It consists of 632 audio event classes organized in a hierarchical ontology, covering a wide range of sounds, including human and animal sounds, musical instruments, genres, and common environmental sounds. The dataset contains approximately 5,790 hours of annotated audio. For our experiments, we used a 98.53-hour randomly sampled subset of the full dataset for training.

**Free Music Archive.** FMA (Defferrard et al., 2016) is a large-scale, open-source dataset of music tracks with clear licensing. We used the "large" subset containing 879.29 hours of audio data with 106,574 30-second tracks sampled at 44.1kHz. The dataset spans multiple genres, including Rock, Electronic, Experimental, Hip-Hop, Folk, and Instrumental, making it suitable for training robust watermarking models across diverse musical styles.

**MusicCaps.** MusicCaps (Agostinelli et al., 2023) is a dataset of 5.5k music-text pairs (9.47 hours total), with high-quality human-written captions describing musical attributes like genre, mood, instruments, and tempo. The dataset was created by having music experts write detailed captions for a subset of 5.5k music clips from AudioSet, with each caption carefully describing the musical content, instrumentation, style, and other sonic characteristics. We used a subset of 9 hours for training.

**VoxPopuli.** VoxPopuli (Wang et al., 2021) is a large-scale multilingual speech corpus collected from European Parliament event recordings between 2009-2020. It contains approximately 400K hours of unlabeled speech data across 23 languages, 1.8K hours of transcribed speech for 16 languages, and 17.3K hours of speech-to-speech interpretation data. For English specifically, it provides 24.1K hours of unlabeled speech data and 543 hours of transcribed speech from 1,313 speakers. The dataset has been widely used for representation learning, semi-supervised learning, and interpretation tasks. We used a 3,016.43-hour subset that covers balanced speech from 23 languages for training.

## B. Discussion on Limitations

While our method demonstrates strong performance across various standard audio transformations, several limitations and opportunities for improvement remain. First, although we achieve robust performance against most common audio edits, our attribution accuracy degrades significantly for challenging operations like speed changes and generative editing. Specifically, as shown in Table 7, XATTNMARK achieves only 4% average attribution accuracy for speed changes across different user setups, though this still represents an improvement over existing baselines, which fail completely. To enhance robustness against speed variations, one promising direction is to incorporate a temporal adjustment layer similar to Westerfeld (2020), which performs black-box optimization to identify and reverse speed changes. We present preliminary work in this direction in App. C.3.1. Additionally, the current solution's attribution robustness against AI-based editing remains an area requiring further improvement.

## C. More Results

### C.1. Subjective Evaluation with MUSHRA

The MUSHRA protocol is a crowdsourced test in which participants rate the quality of various samples on a scale of 0 to 100. The ground truth is provided for reference. We utilized eight randomly sampled speech and music samples from different sources, each lasting 5 seconds. As part of the study, we included a low anchor, which is a low-pass-filtered version at 3.5kHz. Each sample was evaluated by 16 participants, with a post-hoc filtering process to exclude participants who failed to correctly identify more than two hidden reference samples in their scoring, resulting in 12 valid participants at the end. For comparison, the ground truth samples received an average score of around 95, while the low anchor's average score was around 25. As shown in Figure 7, our method achieved a score of approximately 91, closely matching AudioSeal. Among all the methods, AudiowMark is the one that achieves the highest score at around 94. WavMark received a slightly lower score of 88, while TimbreWM scored the lowest, around 80. These results demonstrate that our watermarking approach maintains a perceptual quality that is competitive with state-of-the-art methods.

### C.2. Analysis on the Training Dynamics of Models with Different Architectures

We analyze the training dynamics of different architectures under a controlled experimental setup to better understand their inherent learning capabilities. To isolate architectural effects, we remove most auxiliary losses, retaining only the adversarial loss (weight=1) for all methods. Additionally, we include a waveform-domain $\ell_1$ loss (weight=1) for AudioSeal and our method, as we observed it improves the learning efficiency. Figure 8 illustrates the validation accuracy trajectories and quality for both watermark detection and message-bit decoding during training. The results reveal distinct learning characteristics that highlight the trade-offs between different architectural choices.

WavMark's fully-shared architecture exhibits remarkably rapid initial learning, converging to high accuracy within just a few hundred steps. However, this early success is followed by a concerning degradation in detection accuracy as training

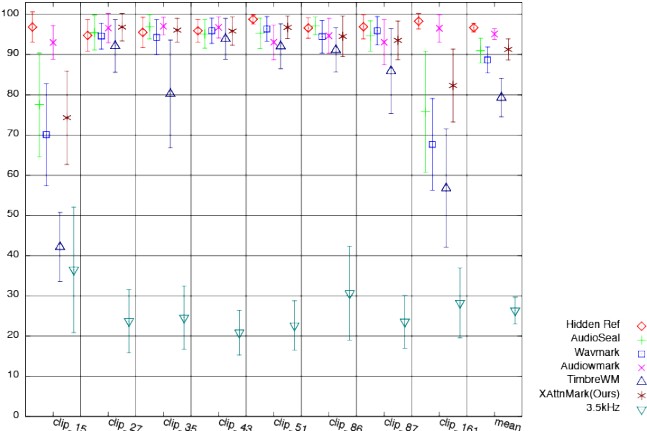

*Figure 7.* MUSHRA subjective listening test results comparing perceptual quality across different watermarking methods. Higher scores indicate better audio quality as rated by human listeners. Our method achieves quality scores comparable to AudioSeal.

progresses. This pattern suggests that while full parameter sharing enables quick initial learning, it may lead to destructive interference between the detection and the decoding tasks during extended training.

AudioSeal's fully-disjoint architecture in general shows slow convergence in both detection and message-bit decoding. After training for 30k steps, it quickly converges to 99% detection accuracy but the learning of message-bit decoding progresses slowly, achieving around 70% message bit accuracy at the end of 50k training steps. Moreover, the perceptual quality of AudioSeal is also degraded more than XATTNMARK and WavMark.

Compared to these, our blended architecture demonstrates steady and comprehensive learning - both detection and decoding accuracies improve gradually, ultimately achieving near-perfect performance only with about 19k training steps. Notably, with only 2k training steps, XATTNMARK can converge to 99% detection accuracy. This great boost in learning efficiency suggests that our partial parameter-sharing strategy and cross-attention mechanism enable more efficient learning of both tasks.

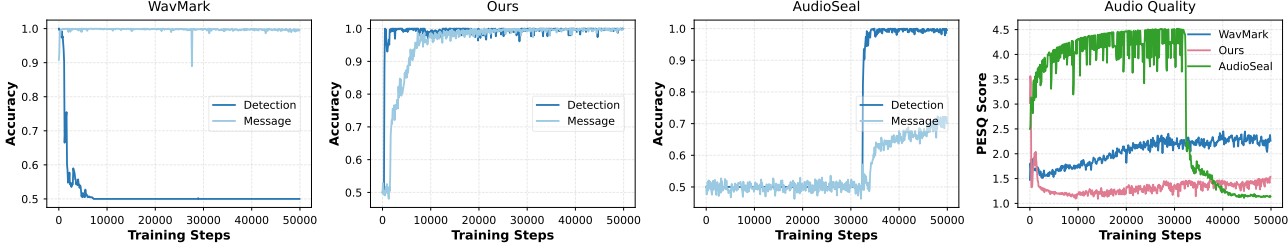

*Figure 8.* Validation accuracy and quality curve of different methods over training steps. Using a blended architecture, XATTNMARK is able to achieve a better balance in terms of learning efficiency in detection, and message-bit decoding and watermark imperceptibility. Compared to our method, WavMark, the fully-shared architecture, suffers from robustness degradation in detection as the training progresses; the fully-disjoint architecture, AudioSeal, suffers from training efficiency issues in learning message-bit decoding.

### C.3. Robustness to Standard Edits

We present the full results summarizing detection and attribution robustness against standard edits in Tables 6 and 7. For detection, our method achieves consistently high performance across all transformations, with an average accuracy of 99.19% and AUC of 99.93%, significantly outperforming all baselines. Even for challenging transformations like speed changes and neural codec compression (EnCodec) where other methods mostly fail (accuracy around 50-60%), our approach maintains exceptional detection performance with accuracies of 99.5% and 96.5%, respectively.

For attribution, the results in Table 7 show that our method maintains high attribution accuracy (92-94% average) across different user pool numbers (and associated messages), demonstrating strong scalability. The performance is particularly

robust for most standard audio transformations like bandpass filtering, boost/duck, echo, and noise addition, achieving near 100% attribution accuracy. While speed changes remain challenging, our method still achieves better performance than baselines which completely fail. This indicates that while our method handles most common audio edits exceptionally well, speed changes continue to be a challenging transformation for reliable attribution. We also explore a potential simple extension of speed reversion to further boost the performance in App. C.3.1.

Our approach shows particularly strong robustness against modern audio codecs - maintaining near-perfect attribution accuracy for both MP3 and AAC compression (98-100%), while baselines like AudioSeal struggle significantly with these transformations (showing accuracies below 60%). This demonstrates our method's effectiveness against state-of-the-art audio compression techniques that are increasingly common in real-world applications. Additionally, our method shows outstanding robustness to cropping operations, maintaining 100% attribution accuracy, while AudioSeal struggles in this task (12% accuracy).

To further clarify our attribution evaluation, particularly regarding the metrics used, we also report the False Attribution Rate (FAR) and compare different definitions of attribution accuracy, aligning with methodologies in AudioSeal (San Roman et al., 2024). In our work, attribution accuracy is defined as *the fraction of correctly attributed samples among those successfully detected as watermarked*. This metric is equivalent to $1 - \text{FAR}$ (False Attribution Rate, as reported in AudioSeal). We adopt this definition because it allows for a decoupled analysis of pure attribution performance from the initial detection performance. In AudioSeal, the attribution accuracy is defined as *the fraction of correctly attributed samples among all samples*, which couples the performance of both detection and attribution. We present the results on both the MusicCaps and VoxPopuli datasets in Table 5, illustrating performance under these specific metrics. The results suggest that in the speech domain, when the user pool size is relatively small, AudioSeal is slightly better than XATTNMARK, and in all the other setups, XATTNMARK consistently outperforms AudioSeal and WavMark.

*Table 5.* Attribution results over edits on MusicCaps (our main setup) and VoxPopuli (align with AudioSeal's setup in San Roman et al. (2024)). We report the false attribution rate (FAR) and two attribution accuracy metrics (Att. and Det.+Att.). The key difference between our definition of attribution accuracy and AudioSeal's is that we isolate the attribution performance from the detection performance. The best score is in **bold**, and the runner-up is underlined.

| Dataset | Metric | Method | N=10 | N=100 | N=1000 | N=10000 |
|---------|--------|--------|------|-------|--------|---------|
| MusicCaps | FAR (%) ↓ | WavMark | 14.67 | 14.92 | 15.18 | 16.07 |
| | | AudioSeal | 18.77 | 25.76 | 36.63 | 30.53 |
| | | XAttnMark | **6.40** | **6.13** | **6.44** | **6.56** |
| | Acc. (Att.) (%) ↑ | WavMark | 85.33 | 85.08 | 84.82 | 83.93 |
| | | AudioSeal | 81.23 | 74.24 | 63.37 | 69.47 |
| | | XAttnMark | **93.60** | **93.87** | **93.56** | **93.44** |
| | Acc. (Det.+Att.) (%) ↑ | WavMark | 79.69 | 80.56 | 79.94 | 79.06 |
| | | AudioSeal | 80.81 | 73.56 | 62.69 | 68.75 |
| | | XAttnMark | **90.50** | **87.19** | **90.75** | **91.81** |
| VoxPopuli | FAR (%) ↓ | WavMark | 13.11 | 13.21 | 13.96 | 15.00 |
| | | AudioSeal | **3.71** | 12.86 | 18.91 | 16.31 |
| | | XAttnMark | 5.03 | **6.01** | **6.25** | **7.70** |
| | Acc. (Att.) (%) ↑ | WavMark | 86.89 | 86.79 | 86.04 | 85.00 |
| | | AudioSeal | **96.29** | 87.14 | 81.09 | 83.69 |
| | | XAttnMark | 94.97 | **93.99** | **93.75** | **92.30** |
| | Acc. (Det.+Att.) (%) ↑ | WavMark | 78.69 | 78.34 | 78.06 | 76.88 |
| | | AudioSeal | 89.50 | 86.60 | 79.83 | 83.45 |
| | | XAttnMark | **92.45** | **92.69** | **92.49** | **91.08** |

### C.3.1. SIMPLE SPEED REVERSION LAYER WITH BLACK-BOX GRID SEARCH

We explore a post-hoc simple speed reversion layer with a black-box grid search to further boost the robustness against speed changes. The speed reversion layer is added on top of the watermarking model. The main idea is to reverse the speed

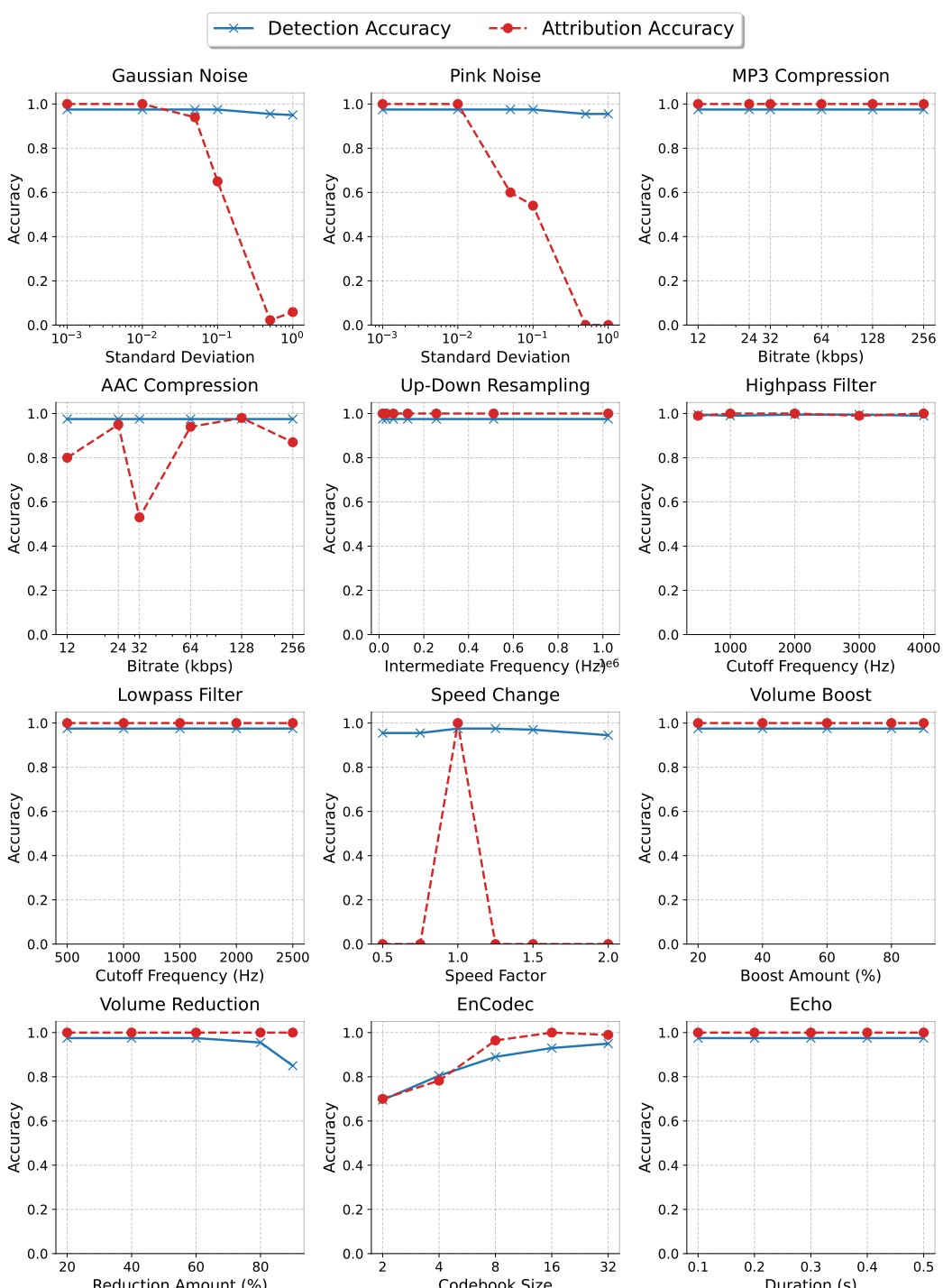

*Figure 9.* Detection and Attribution Accuracy of our method on augmented samples across different augmentation strengths.

*Table 6.* **The full detection results** of XATTNMARK across different standard audio edits on the MusicCaps dataset. Acc. (TPR/FPR) is the accuracy (and TPR/FPR) obtained for the optimal threshold from Youden's Index (Youden, 1950) on a balanced validation set.

| Edit | AudiowMark Acc. (TPR/FPR) | AUC | WavMark Acc. (TPR/FPR) | AUC | TimbreWM Acc. (TPR/FPR) | AUC | AudioSeal Acc. (TPR/FPR) | AUC | XATTNMARK Acc. (TPR/FPR) | AUC |
|---|---|---|---|---|---|---|---|---|---|---|
| Identity | 1.00 (1.00/0.00) | 1.00 | 1.00 (1.00/0.00) | 1.00 | 0.995 (0.99/0.00) | 1.00 | 1.00 (0.99/0.00) | 1.00 | 0.995 (0.99/0.00) | 1.00 |
| Bandpass | 1.00 (1.00/0.00) | 1.00 | 1.00 (1.00/0.00) | 1.00 | 0.985 (0.97/0.00) | 1.00 | 1.00 (0.99/0.00) | 1.00 | 0.995 (0.99/0.00) | 1.00 |
| Boost | 1.00 (1.00/0.00) | 1.00 | 1.00 (1.00/0.00) | 1.00 | 0.98 (0.96/0.00) | 0.9998 | 1.00 (0.99/0.00) | 1.00 | 0.995 (0.99/0.00) | 1.00 |
| Duck | 1.00 (1.00/0.00) | 1.00 | 0.995 (0.99/0.00) | 0.995 | 0.97 (0.94/0.00) | 0.9999 | 1.00 (0.99/0.00) | 1.00 | 0.995 (0.99/0.00) | 1.00 |
| Echo | 1.00 (1.00/0.00) | 1.00 | 1.00 (1.00/0.00) | 1.00 | 0.945 (0.89/0.00) | 0.9965 | 1.00 (0.99/0.00) | 1.00 | 0.995 (0.99/0.00) | 1.00 |
| Highpass | 1.00 (1.00/0.00) | 1.00 | 0.95 (0.90/0.00) | 0.95 | 0.985 (0.97/0.00) | 1.00 | 1.00 (0.99/0.00) | 1.00 | 0.995 (0.99/0.00) | 1.00 |
| Lowpass | 1.00 (1.00/0.00) | 1.00 | 1.00 (1.00/0.00) | 1.00 | 0.985 (0.97/0.00) | 0.9986 | 1.00 (0.99/0.00) | 1.00 | 0.995 (0.99/0.00) | 1.00 |
| MP3 | 0.94 (0.88/0.00) | 0.94 | 0.805 (0.61/0.00) | 0.805 | 0.95 (0.91/0.01) | 0.9971 | 1.00 (0.99/0.00) | 1.00 | 0.995 (0.99/0.00) | 1.00 |
| Pink Noise | 1.00 (1.00/0.00) | 1.00 | 1.00 (1.00/0.00) | 1.00 | 0.995 (0.99/0.00) | 1.00 | 1.00 (0.99/0.00) | 1.00 | 0.995 (0.99/0.00) | 1.00 |
| White Noise | 1.00 (1.00/0.00) | 1.00 | 1.00 (1.00/0.00) | 1.00 | 0.975 (0.95/0.00) | 0.9999 | 1.00 (0.99/0.00) | 1.00 | 0.995 (0.99/0.00) | 1.00 |
| Smooth | 1.00 (1.00/0.00) | 1.00 | 0.96 (0.92/0.00) | 0.96 | 0.98 (0.96/0.00) | 0.9976 | 1.00 (0.99/0.00) | 1.00 | 0.995 (0.99/0.00) | 1.00 |
| Speed | 0.50 (0.00/0.00) | 0.50 | 0.50 (0.00/0.00) | 0.50 | 0.515 (0.07/0.04) | 0.4812 | 0.61 (0.36/0.15) | 0.61 | 0.995 (0.99/0.00) | 1.00 |
| Resample | 1.00 (1.00/0.00) | 1.00 | 1.00 (1.00/0.00) | 1.00 | 0.98 (0.96/0.00) | 1.00 | 1.00 (0.99/0.00) | 1.00 | 0.995 (0.99/0.00) | 1.00 |
| AAC | 1.00 (1.00/0.00) | 1.00 | 1.00 (1.00/0.00) | 1.00 | 0.975 (0.95/0.00) | 0.9999 | 1.00 (0.99/0.00) | 1.00 | 0.995 (0.99/0.00) | 1.00 |
| EnCodec (nq=16) | 0.50 (0.00/0.00) | 0.50 | 0.805 (0.61/0.00) | 0.805 | 0.625 (0.50/0.25) | 0.7092 | 1.00 (0.99/0.00) | 1.00 | 0.965 (0.93/0.00) | 0.991 |
| Crop | 0.965 (0.93/0.00) | 0.965 | 0.995 (0.99/0.00) | 0.995 | 0.96 (0.92/0.00) | 0.9918 | 1.00 (0.99/0.00) | 1.00 | 0.975 (0.98/0.03) | 0.997 |
| **Average** | 0.9294 (0.8588/0.00) | 0.9294 | 0.9181 (0.8363/0.00) | 0.9181 | 0.925 (0.8688/0.0188) | 0.9482 | 0.9706 (0.95/0.01) | 0.9757 | 0.9919 (0.9856/0.0019) | 0.9993 |

*Table 7.* The full attribution results for all methods, and associated overall average performance, across all transformations on the MusicCaps dataset.

| Transformation | AudiowMark N=100 | 1000 | 10000 | Avg | WavMark 100 | 1000 | 10000 | Avg | TimbreWM 100 | 1000 | 10000 | Avg | AudioSeal 100 | 1000 | 10000 | Avg | XATTNMARK 100 | 1000 | 10000 | Avg |
|---|---|---|---|---|---|---|---|---|---|---|---|---|---|---|---|---|---|---|---|---|
| **Identity** | 1.00 | 1.00 | 1.00 | 1.00 | 1.00 | 1.00 | 1.00 | 1.00 | 0.96 | 0.93 | 0.86 | 0.93 | 0.75 | 0.50 | 0.50 | 0.69 | 1.00 | 1.00 | 1.00 | 1.00 |
| **EnCodec** | 0.00 | 0.00 | 0.00 | 0.00 | 0.00 | 0.00 | 0.00 | 0.00 | 0.03 | 0.00 | 0.00 | 0.07 | 0.50 | 0.00 | 0.00 | 0.31 | 1.00 | 0.99 | 0.99 | 0.99 |
| **AAC** | 1.00 | 1.00 | 1.00 | 1.00 | 1.00 | 1.00 | 1.00 | 1.00 | 0.94 | 0.90 | 0.85 | 0.90 | 0.50 | 0.00 | 0.00 | 0.12 | 0.99 | 0.96 | 0.69 | 0.88 |
| **Bandpass** | 1.00 | 1.00 | 1.00 | 1.00 | 1.00 | 1.00 | 1.00 | 1.00 | 0.92 | 0.92 | 0.89 | 0.93 | 0.50 | 0.25 | 0.00 | 0.31 | 0.99 | 0.99 | 0.99 | 0.99 |
| **Boost** | 1.00 | 1.00 | 1.00 | 1.00 | 1.00 | 1.00 | 1.00 | 1.00 | 0.91 | 0.91 | 0.87 | 0.91 | 0.25 | 0.50 | 0.25 | 0.50 | 1.00 | 1.00 | 1.00 | 1.00 |
| **Duck** | 1.00 | 1.00 | 1.00 | 1.00 | 1.00 | 1.00 | 1.00 | 1.00 | 0.93 | 0.85 | 0.84 | 0.89 | 0.50 | 0.50 | 0.25 | 0.56 | 1.00 | 1.00 | 1.00 | 1.00 |
| **Echo** | 1.00 | 1.00 | 1.00 | 1.00 | 1.00 | 1.00 | 1.00 | 1.00 | 0.92 | 0.92 | 0.87 | 0.91 | 0.25 | 0.25 | 0.25 | 0.38 | 1.00 | 0.98 | 1.00 | 0.99 |
| **Highpass** | 1.00 | 1.00 | 1.00 | 1.00 | 1.00 | 1.00 | 1.00 | 1.00 | 0.96 | 0.92 | 0.87 | 0.93 | 0.50 | 0.25 | 0.00 | 0.31 | 1.00 | 1.00 | 1.00 | 1.00 |
| **Lowpass** | 1.00 | 1.00 | 1.00 | 1.00 | 1.00 | 1.00 | 1.00 | 1.00 | 0.93 | 0.86 | 0.85 | 0.90 | 0.50 | 0.50 | 0.50 | 0.56 | 1.00 | 1.00 | 1.00 | 1.00 |
| **MP3** | 1.00 | 1.00 | 1.00 | 1.00 | 1.00 | 1.00 | 1.00 | 1.00 | 0.85 | 0.85 | 0.86 | 0.86 | 0.50 | 0.50 | 0.00 | 0.38 | 1.00 | 1.00 | 1.00 | 1.00 |
| **Pink Noise** | 1.00 | 1.00 | 1.00 | 1.00 | 1.00 | 1.00 | 1.00 | 1.00 | 0.93 | 0.91 | 0.87 | 0.92 | 0.75 | 0.75 | 0.75 | 0.75 | 1.00 | 1.00 | 1.00 | 1.00 |
| **White Noise** | 1.00 | 1.00 | 1.00 | 1.00 | 1.00 | 1.00 | 1.00 | 1.00 | 0.88 | 0.89 | 0.87 | 0.90 | 0.50 | 0.75 | 0.50 | 0.56 | 1.00 | 1.00 | 1.00 | 1.00 |
| **Smooth** | 1.00 | 1.00 | 1.00 | 1.00 | 1.00 | 1.00 | 0.75 | 0.88 | 0.75 | 0.86 | 0.69 | 0.80 | 0.00 | 0.25 | 0.25 | 0.19 | 1.00 | 1.00 | 0.99 | 1.00 |
| **Speed** | 0.00 | 0.00 | 0.00 | 0.00 | 0.00 | 0.00 | 0.00 | 0.00 | 0.20 | 0.04 | 0.19 | 0.18 | 0.00 | 0.00 | 0.00 | 0.00 | 0.00 | 0.00 | 0.08 | 0.03 |
| **Resample** | 1.00 | 1.00 | 1.00 | 1.00 | 1.00 | 1.00 | 1.00 | 1.00 | 0.93 | 0.92 | 0.87 | 0.92 | 0.75 | 0.25 | 0.25 | 0.56 | 1.00 | 1.00 | 1.00 | 1.00 |
| **Crop** | 1.00 | 1.00 | 1.00 | 1.00 | 1.00 | 1.00 | 1.00 | 1.00 | 0.93 | 0.76 | 0.80 | 0.85 | 0.00 | 0.00 | 0.00 | 0.12 | 1.00 | 0.99 | 1.00 | 1.00 |
| **Avg** | 0.88 | 0.88 | 0.88 | 0.88 | 0.88 | 0.88 | 0.86 | 0.87 | 0.81 | 0.78 | 0.75 | 0.80 | 0.42 | 0.33 | 0.22 | 0.39 | 0.94 | 0.93 | 0.92 | 0.93 |

change by searching for the optimal speed factor with the guidance of the watermarking model. We observe that on the unwatermarked audio, changing the speed can not adversarially increase the detection accuracy, while on the watermarked audio, speed changing will significantly affect the watermark message decoding pattern. We observe that the watermarking model predicts more stable message bits for each 1-second chunk when the speed remains unchanged. However, after applying a speed change, the model predicts more diverse message bits with lower confidence for each chunk. To leverage this insight, given the original speed change space $[\gamma_{\min}, \gamma_{\max}]$ and the subsequent search space $[\frac{1}{\gamma_{\max}}, \frac{1}{\gamma_{\min}}]$, we design the score function $r(\alpha)$ as the average of the watermarking model's mean detection score over each chunk $p(\alpha)$, and the average standard deviation of the predicted message bits $\bar{s}_{m(\alpha)}$. Formally, this is expressed as $r(\alpha) = \frac{1}{L} \sum_{i=1}^{L} (p_i(\alpha) + \bar{s}_{m,i}(\alpha))$, where $L$ is the number of chunks, $\alpha$ is the speed factor, $p_i(\alpha)$ is the detection score for chunk $i$, and $\bar{s}_{m,i}(\alpha)$ is the average standard deviation of the predicted message bits for chunk $i$. Then we conduct a two-round linear grid search to find the optimal speed factor $\alpha$ that maximizes the score function $r(\alpha)$, with first-round step size $S_1$ and second-round step size $S_2$. The speed change parameter $\gamma_{\min}$ and $\gamma_{\max}$ are set to 0.8 and 1.25, respectively. We conduct experiments on both watermarked and unwatermarked audio to investigate whether speed reversion might cause false positives. For the watermarked case, we report the attribution success rate/detection time, and for the unwatermarked case, we report the FPR. As shown by the results in Table 8, with a more fine-grained search space, the speed reversion layer can further improve the robustness against speed changes while maintaining the overall detection efficiency. Meanwhile, for unwatermarked audio, applying the speed reversion layer does not falsely increase the probability of being classified as watermarked by the detection model.

*Table 8.* Comparison of speed reversion layer performance. We report both the attribution success rate and the average detection time (in seconds) for different user pool sizes $N$. The speed reversion layer significantly improves robustness against speed changes while maintaining efficient detection time.

| #User | w/o Search | | w/ Search ($S_1 = 0.1, S_2 = 0.01$) | | | w/ Search ($S_1 = 0.01, S_2 = 0.001$) | | |
|---|---|---|---|---|---|---|---|---|
| | Success Rate | Det. Time (s) | Success Rate | Det. Time (s) | FPR | Success Rate | Det. Time (s) | FPR |
| $N = 2$ | 0.391 | 0.005 | 0.537 | 0.220 | 0.01 | **1.000** | 0.653 | 0.00 |
| $N = 10$ | 0.069 | 0.006 | 0.237 | 0.229 | 0.00 | **1.000** | 0.583 | 0.01 |
| $N = 100$ | 0.222 | 0.006 | 0.612 | 0.230 | 0.00 | **0.990** | 0.545 | 0.00 |
| $N = 1000$ | 0.000 | 0.007 | 0.163 | 0.230 | 0.00 | **1.000** | 0.497 | 0.00 |

## C.4. Robustness to Generative Edits

We use the DDIM-inversion-based editing approach of ZETA (Text-Based Audio Editing) introduced in Manor & Michaeli (2024) to manipulate the audio with text prompts. We briefly summarize the ZETA editing process in the next section. For more details on this editing process, please refer to Manor & Michaeli (2024).

**ZETA.** Given the original signal $x_0$ (waveform or latent space), this approach first conduct guided forward process $x_t = \sqrt{\bar{\alpha}_t}x_0 + \sqrt{1 - \bar{\alpha}_t}\tilde{e}_t$, $t = 1, \ldots, T$, where $\bar{\alpha}_t$ is the cumulative variance schedule, and $\tilde{e}_t$ is the independent Gaussian noise. Then a sequence of noise vectors that capture the main characteristics of the source audio is extracted using $z_t = (x_{t-1} - \mu_t(x_t))/\sigma_t$, $t = T, \ldots, 1$, where $\mu_t(\cdot)$ is a denoising neural network trained to predict the mean component of the noise added at time $t$. Note that a source prompt $p_{src}$ can also be used at the conditioning layer of the denoising neural network to guide the forward diffusion process. During the editing process, given a target prompt $p_{tgt}$ and a noisy state $x_T$, a guided diffusion backward process is conducted via $x_{t-1} = \mu_t(x_t) + \sigma_t z_t, t = T, \ldots, 1$, where the noise vector sequence $z_t$ are the ones extracted from the previous forward process. The target prompt $p_{tgt}$ is embedded and applied on the cross-attention part that affects the editing trajectory.

We present the full results of robustness against generative edits in Table 9, where we evaluate our method against two state-of-the-art audio generation models: Stable Audio (Evans et al., 2024) and AudioLDM2 (Liu et al., 2024b). Different forward diffusion steps $T$ are used to create varying strengths of audio manipulation, ranging from 10 to 110. The source prompt $p_{src}$ is set to "A recording of music", and the target prompt $p_{tgt}$ is set to "A recording of EDM music with strong rhythm". We follow the default settings in the original paper for other hyperparameters.

For Stable Audio, our method achieves consistently high accuracy (0.81-0.94) and AUC scores (0.89-0.99) across different $T$ values. In contrast, AudioSeal shows more variable performance: while it achieves decent accuracy (0.69-0.78) for lower $T$ values (10-30), its performance degrades significantly as $T$ increases, with accuracy dropping to around 0.59 and AUC falling to around 0.50-0.53 for higher $T$ values. For AudioLDM2, our approach maintains strong performance with an accuracy of around 0.94 and an AUC above 0.93 across all diffusion steps. AudioSeal again struggles with higher $T$ values, showing consistent accuracy around 0.59 but poor AUC scores (0.33-0.53).

Furthermore, we also compare against Timbre Watermark (Liu et al., 2023a), a frequency-based watermarking method that has demonstrated state-of-the-art robustness against voice cloning training pipelines. While this approach could potentially handle generative edits, which also modify audio significantly, our experiments show that it performs poorly on this task. As shown in Table 9, Timbre Watermark achieves only random-guess level performance (accuracy around 0.50) for both Stable Audio and AudioLDM2 across different $T$ values, significantly underperforming both AudioSeal and our method.

## C.5. Generalization Across Standard Editing with Different Strengths

We present the generalization of XATTNMARK against transformations of different strengths in Figure 9. Our method demonstrates remarkable robustness across a wide spectrum of audio transformations with varying intensities. For **lossy compression operations**, EnCodec shows detection accuracy of 69.5-95.0% and attribution rates of 70.0-100% depending on bitrate. AAC maintains strong detection (97.5%) across all bitrates with attribution ranging from 53.0-98.0%. MP3 compression achieves consistent detection (97.5%) and perfect attribution (100%) across all bitrates.

For **frequency-domain operations**, our watermark remains highly effective. When filtering out low frequencies (highpass filtering), we maintain excellent performance with 99.0-99.5% detection accuracy and 99.0-100% attribution accuracy, even

*Table 9.* We demonstrate that our watermarking approach can generalize better to the unseen, challenging generative edits setting. **Detection Results** for two generative edits applied with different forward $t$ steps.

| Editing Model | $T$ | AudioSeal | | TimbreWatermark | | XATTNMARK | |
|---|---|---|---|---|---|---|---|
| | | Acc. (TPR/FPR) | AUC | Acc. (TPR/FPR) | AUC | Acc. (TPR/FPR) | AUC |
| **Stable Audio** | 10 | 0.6875 (0.625/0.25) | 0.7227 | 0.4688 (0.8125/0.875) | 0.4219 | 0.9375 (0.9375/0.0625) | 0.9883 |
| | 30 | 0.7813 (0.875/0.3125) | 0.8125 | 0.4063 (0.5625/0.75) | 0.3789 | 0.8125 (0.75/0.125) | 0.8906 |
| | 50 | 0.5938 (0.875/0.75) | 0.3398 | 0.4688 (0.0625/0.125) | 0.4863 | 0.9063 (0.875/0.0625) | 0.9688 |
| | 70 | 0.5938 (0.75/0.5625) | 0.5313 | 0.5313 (0.1875/0.125) | 0.5547 | 0.9063 (0.875/0.0625) | 0.9648 |
| | 90 | 0.5938 (0.875/0.6875) | 0.5469 | 0.5000 (0.5000/0.5000) | 0.5996 | 0.9063 (0.875/0.0625) | 0.9688 |
| | 110 | 0.5938 (0.9375/0.75) | 0.5039 | 0.5313 (0.5625/0.5000) | 0.5488 | 0.9063 (0.875/0.0625) | 0.9570 |
| **AudioLDM2-Music** | 10 | 0.5938 (0.9375/0.75) | 0.5273 | 0.5313 (0.8125/0.75) | 0.4961 | 0.9375 (0.9375/0.0625) | 0.9492 |
| | 30 | 0.5938 (0.9375/0.75) | 0.3945 | 0.4375 (0.0/0.125) | 0.4570 | 0.9375 (0.9375/0.0625) | 0.9414 |
| | 50 | 0.56 (0.88/0.75) | 0.34 | 0.5313 (0.5625/0.5000) | 0.5645 | 0.9063 (0.875/0.0625) | 0.9336 |
| | 70 | 0.5938 (0.9375/0.75) | 0.3477 | 0.5000 (0.125/0.125) | 0.5703 | 0.9375 (0.9375/0.0625) | 0.9453 |
| | 90 | 0.5938 (0.9375/0.75) | 0.3945 | 0.5000 (0.5000/0.5000) | 0.5723 | 0.9375 (0.9375/0.0625) | 0.9414 |
| | 110 | 0.5938 (0.9375/0.75) | 0.3359 | 0.5000 (0.0/0.0) | 0.5137 | 0.9375 (0.9375/0.0625) | 0.9414 |

as we vary the cutoff frequency from 500Hz to 4000Hz. Similarly, when filtering out high frequencies (lowpass filtering with cutoffs between 500-2500Hz), we achieve 97.5% detection accuracy while maintaining 100% attribution accuracy. For volume changes, our method handles both increases and reductions well - maintaining 97.5% detection and 100% attribution for volume boosts up to 90%, and only showing slight degradation (85.0% detection) when reducing volume to 10% of original while still maintaining 100% attribution.

For **additive noise**, both Gaussian and pink noise maintain 97.5% detection accuracy up to 0.1 standard deviations, though attribution degrades from 100% at low noise levels ($\leq 0.01$) to 0% at high intensities ($\geq 0.5$). **Speed changes** achieve 94.5-97.5% detection but, as expected, only maintain attribution at normal speed (1.0x). For **resampling operations** across frequencies from 16kHz to 1024kHz, detection remains strong (97.5%) with perfect attribution (100%). Echo effects across different delay times (0.1-0.5) consistently achieve 97.5% detection and 100% attribution, demonstrating the watermark's comprehensive resilience to common audio transformations.

### C.6. Generalization Across Different Datasets

To study how XATTNMARK generalizes across different domains and datasets, we conduct in-distribution and out-of-distribution tests on multiple audio datasets. We randomly sample 100 audio clips with a 5s duration from each dataset. The user size $N$ is set to 1000.

For in-distribution evaluation (Table 10), we test on FMA-Large (music), LibriSpeech (speech), VoxPopuli-10K (speech), and AudioSet (general audio). Our method achieves consistently high detection accuracy (96-99%) and attribution accuracy (87-93%) across all datasets and editing operations. The only exception is speed change, where attribution fails as expected since it fundamentally alters the temporal structure that watermarks rely on. We also observe slightly lower performance on EnCodec compression for FMA-Large and AudioSet, where detection accuracy drops to 72-81% and attribution accuracy to 70-97%, reflecting the challenge of handling highly compressed audio. Notably, the average detection accuracy across all datasets remains strong at 96-99%, with attribution accuracy ranging from 87-93%.

For out-of-distribution generalization (Table 11), we evaluate on four unseen datasets: AudioMarkBench (Liu et al., 2024a) (which is the speech subset from Common Voice (Ardila et al., 2020)), ASVspoof (speech) (Liu et al., 2023b), MusicGen (AI-generated music) (Copet et al., 2024), and MUSDB18 (music) (Rafii et al., 2017). Our method maintains robust performance on these challenging OOD datasets, with detection accuracy ranging from 93-99% and attribution accuracy from 86-94%. The speech datasets (AudioMarkBench and ASVspoof) show slightly lower but still strong detection rates of 93-97%, while music datasets (MusicGen and MUSDB18) achieve near-perfect detection at 98-99%. Attribution performance remains consistently high across all OOD datasets, except for speed changes, which consistently fail as expected. Notably, the average attribution accuracy for OOD datasets ranges from 86-94%, demonstrating the robustness of our watermarking approach across diverse audio domains.

These results highlight the strong generalization capabilities of our method, both in-distribution and out-of-distribution. The

*Figure 10.* The detection and attribution accuracy of XATTNMARK across different audio duration and watermark strength $\alpha$ on MusicCaps. The performance is averaged over all the standard audio editing. The attribution pool size is set to $\{100, 1000, 10000\}$.

consistent performance across datasets and editing operations, with only expected failures (e.g., speed changes and extreme compression), underscores the practical applicability of our watermarking system in real-world scenarios where diverse audio content and editing operations are encountered.

### C.7. Performance across the different audio duration and watermark strength $\alpha$

To investigate the performance of our method across different audio durations and watermark strength $\alpha$, we conduct experiments with audio duration ranging from 1s to 10s and watermark strength $\alpha$ ranging from 0.0 to 1.5. As shown in Figure 10, we analyze the impact of these parameters on detection and attribution accuracy.

For duration analysis, we observe that the detection accuracy remains robust (98.6–99.3%) across all tested durations from 1–10s, demonstrating our method's effectiveness even with short audio clips. Attribution accuracy shows greater sensitivity, improving from 81.2% at 1s to a peak of 93.0% at 5s, then stabilizing at 91.5–92.6% for longer durations. This suggests 5s segments provide the optimal trade-off between practical usability and attribution performance, though reliable detection (>98.5%) is achievable with clips as short as 1s.

For watermark strength analysis, we observe three distinct operational regimes: i) weak watermarks ($\alpha \leq 0.3$), ii) moderate watermarks ($0.4 \leq \alpha \leq 0.7$), and iii) strong watermarks ($\alpha \geq 0.8$); achieve asymptotic performance with detection >98.3% and attribution >92.3%, showing diminishing returns beyond $\alpha = 1.0$.

Our default $\alpha = 1.0$ configuration achieves 98.5% detection and 92.7% attribution accuracy. While higher $\alpha$ values (up to 1.5) marginally improve detection to 98.9%, attribution plateaus at 93.1–93.2%, suggesting an upper bound for useful watermark strength. Notably, the complete absence of watermarking ($\alpha = 0$) yields chance-level detection (55.7%) and zero attribution capability, confirming the system's dependence on the embedded signal.

### C.8. Watermark Residual Visualization

To better understand the characteristics of different watermarking methods, we visualize the watermark residuals in Figure 11. The visualization includes waveform, spectrogram, and mel-spectrogram representations for the original audio and four watermarking approaches (XATTNMARK, AudioSeal, WavMark, and AudiowMark). Our analysis reveals several key insights: (1) Our method introduces watermarks with the lowest energy, making them less perceptible in both time and frequency domains. (2) The spectral patterns of our watermarks appear more stealthy and natural compared to other methods. (3) Unlike WavMark, which shows discrete interruptions, our watermarking pattern exhibits better continuity, contributing to enhanced robustness against audio transformations while preserving imperceptibility.

### C.9. Visualization of TF-weighted Penalty Distribution.

**Comparison with TF-loudness loss in AudioSeal.** AudioSeal adopts a coarse approach to implement auditory masking, using a naive loudness difference between the noise and the original audio. The main idea is that high loudness regions should mask lower loudness regions in the watermark residual. A positive penalty is assigned when the residual loudness is higher than that of the original audio for each tile.

Specifically, the waveform is directly divided into $B \times W$ regions, where each region shares the same weighting term. Here, $B$ represents the number of frequency bands for splitting, and $W$ denotes the number of time tiles. The loudness difference between the watermark residual and the original waveform is then computed as $l_b^w = \text{Loudness}(\delta_b^w) - \text{Loudness}(x_b^w)$,

*Table 10.* The performance of detection and attribution of our method on different in-distribution datasets over different editing operations.

| Dataset | FMA-Large | | LibriSpeech | | VoxPopuli-10K | | AudioSet | |
|---|---|---|---|---|---|---|---|---|
| Edit | Det. (TPR/FPR) | Att. | Det. (TPR/FPR) | Att. | Det. (TPR/FPR) | Att. | Det. (TPR/FPR) | Att. |
| Identity | 0.995 (0.990/0.000) | 1.000 | 0.995 (0.990/0.000) | 1.000 | 0.997 (0.994/0.000) | 1.000 | 0.990 (0.980/0.000) | 0.970 |
| Bandpass | 0.995 (0.990/0.000) | 0.980 | 0.995 (0.990/0.000) | 1.000 | 0.997 (0.994/0.000) | 1.000 | 0.995 (0.990/0.000) | 0.990 |
| Boost | 0.995 (0.990/0.000) | 0.990 | 0.995 (0.990/0.000) | 1.000 | 0.997 (0.994/0.000) | 1.000 | 0.985 (0.980/0.010) | 0.970 |
| Duck | 0.995 (0.990/0.000) | 1.000 | 0.995 (0.990/0.000) | 1.000 | 0.997 (0.994/0.000) | 1.000 | 0.995 (0.990/0.000) | 0.980 |
| Echo | 0.990 (0.980/0.000) | 1.000 | 0.995 (0.990/0.000) | 1.000 | 0.997 (0.994/0.000) | 1.000 | 0.985 (0.970/0.000) | 0.960 |
| Highpass | 0.995 (0.990/0.000) | 0.990 | 0.995 (0.990/0.000) | 1.000 | 0.997 (0.994/0.000) | 1.000 | 0.995 (0.990/0.000) | 1.000 |
| Lowpass | 0.965 (0.950/0.020) | 1.000 | 0.995 (0.990/0.000) | 1.000 | 0.994 (0.994/0.006) | 1.000 | 0.955 (0.950/0.040) | 0.969 |
| MP3 | 0.975 (0.970/0.020) | 0.990 | 0.995 (0.990/0.000) | 1.000 | 0.994 (0.994/0.006) | 1.000 | 0.955 (0.960/0.050) | 0.947 |
| Pink Noise | 0.995 (0.990/0.000) | 0.990 | 0.995 (0.990/0.000) | 0.920 | 0.997 (0.994/0.000) | 0.918 | 0.995 (0.990/0.000) | 0.940 |
| White Noise | 0.995 (0.990/0.000) | 1.000 | 0.995 (0.990/0.000) | 1.000 | 0.997 (0.994/0.000) | 1.000 | 0.995 (0.990/0.000) | 0.980 |
| Smooth | 0.990 (0.980/0.000) | 1.000 | 0.995 (0.990/0.000) | 1.000 | 0.997 (0.994/0.000) | 1.000 | 0.985 (0.990/0.020) | 0.927 |
| Speed | 0.905 (0.830/0.020) | 0.000 | 0.995 (0.990/0.000) | 0.000 | 0.994 (0.994/0.006) | 0.000 | 0.935 (0.920/0.050) | 0.000 |
| Resample | 0.970 (0.960/0.020) | 1.000 | 0.995 (0.990/0.000) | 1.000 | 0.994 (0.987/0.000) | 1.000 | 0.960 (0.940/0.020) | 0.949 |
| AAC | 0.990 (0.980/0.000) | 0.960 | 0.995 (0.990/0.000) | 1.000 | 0.997 (0.994/0.000) | 1.000 | 0.980 (0.980/0.020) | 0.950 |
| EnCodec (nq=16) | 0.720 (0.460/0.020) | 0.970 | 0.810 (0.730/0.110) | 0.000 | 0.956 (0.943/0.031) | 1.000 | 0.735 (0.620/0.150) | 0.706 |
| Crop | 0.995 (0.990/0.000) | 0.947 | 0.995 (0.990/0.000) | 1.000 | 0.997 (0.994/0.000) | 0.992 | 0.995 (0.990/0.000) | 0.964 |
| **Average** | 0.967 (0.939/0.006) | 0.926 | 0.983 (0.974/0.007) | 0.870 | 0.994 (0.990/0.003) | 0.932 | 0.965 (0.952/0.023) | 0.888 |

*Table 11.* The performance of detection and attribution of our method on different out-of-distribution datasets (unseen in training) over different editing operations.

| Dataset | AudioMarkBench | | ASVspoof | | MusicGen | | MUSDB18 | |
|---|---|---|---|---|---|---|---|---|
| Edit | Det. (TPR/FPR) | Att. | Det. (TPR/FPR) | Att. | Det. (TPR/FPR) | Att. | Det. (TPR/FPR) | Att. |
| Identity | 0.938 (0.875/0.000) | 1.000 | 0.967 (0.933/0.000) | 1.000 | 0.995 (0.990/0.000) | 0.917 | 0.996 (0.991/0.000) | 1.000 |
| Bandpass | 0.938 (0.875/0.000) | 1.000 | 0.967 (0.933/0.000) | 1.000 | 0.995 (0.990/0.000) | 1.000 | 0.996 (0.991/0.000) | 1.000 |
| Boost | 0.938 (0.875/0.000) | 1.000 | 0.967 (0.933/0.000) | 1.000 | 0.990 (0.979/0.000) | 0.969 | 0.996 (0.991/0.000) | 1.000 |
| Duck | 0.938 (0.875/0.000) | 1.000 | 0.967 (0.933/0.000) | 1.000 | 0.990 (0.979/0.000) | 0.969 | 0.996 (0.991/0.000) | 1.000 |
| Echo | 0.938 (0.875/0.000) | 1.000 | 0.967 (0.933/0.000) | 1.000 | 0.984 (0.979/0.010) | 0.938 | 0.996 (0.991/0.000) | 1.000 |
| Highpass | 0.938 (0.875/0.000) | 1.000 | 0.967 (0.933/0.000) | 1.000 | 0.995 (0.990/0.000) | 1.000 | 0.996 (0.991/0.000) | 1.000 |
| Lowpass | 0.938 (0.875/0.000) | 1.000 | 0.967 (0.933/0.000) | 0.933 | 0.969 (0.979/0.042) | 0.926 | 0.991 (0.991/0.009) | 1.000 |
| MP3 | 0.938 (0.875/0.000) | 1.000 | 0.933 (0.867/0.000) | 0.867 | 0.969 (0.969/0.031) | 0.926 | 0.991 (0.983/0.000) | 1.000 |
| Pink Noise | 0.938 (0.875/0.000) | 1.000 | 0.967 (0.933/0.000) | 1.000 | 0.990 (0.979/0.000) | 0.979 | 0.996 (0.991/0.000) | 1.000 |
| White Noise | 0.938 (0.875/0.000) | 1.000 | 0.967 (0.933/0.000) | 1.000 | 0.990 (0.979/0.000) | 0.896 | 0.996 (0.991/0.000) | 1.000 |
| Smooth | 0.938 (0.875/0.000) | 1.000 | 0.967 (0.933/0.000) | 0.867 | 0.974 (0.979/0.031) | 0.895 | 0.996 (0.991/0.000) | 1.000 |
| Speed | 0.938 (0.875/0.000) | 0.000 | 0.967 (0.933/0.000) | 0.000 | 0.911 (0.896/0.073) | 0.000 | 0.987 (0.983/0.009) | 0.000 |
| Resample | 0.938 (0.875/0.000) | 1.000 | 0.967 (0.933/0.000) | 0.867 | 0.974 (0.979/0.031) | 0.860 | 0.991 (0.991/0.009) | 1.000 |
| AAC | 0.938 (0.875/0.000) | 1.000 | 0.967 (0.933/0.000) | 1.000 | 0.979 (0.990/0.031) | 0.781 | 0.996 (0.991/0.000) | 1.000 |
| EnCodec (nq=16) | 0.875 (0.750/0.000) | 1.000 | 0.867 (0.867/0.133) | 1.000 | 0.885 (0.875/0.104) | 0.833 | 0.957 (0.957/0.043) | 1.000 |
| Crop | 0.938 (0.875/0.000) | 0.000 | 0.967 (0.933/0.000) | 1.000 | 0.891 (0.875/0.094) | 0.871 | 0.905 (0.888/0.078) | 0.988 |
| **Average** | 0.934 (0.867/0.000) | 0.875 | 0.958 (0.925/0.008) | 0.908 | 0.967 (0.963/0.028) | 0.860 | 0.986 (0.982/0.009) | 0.937 |

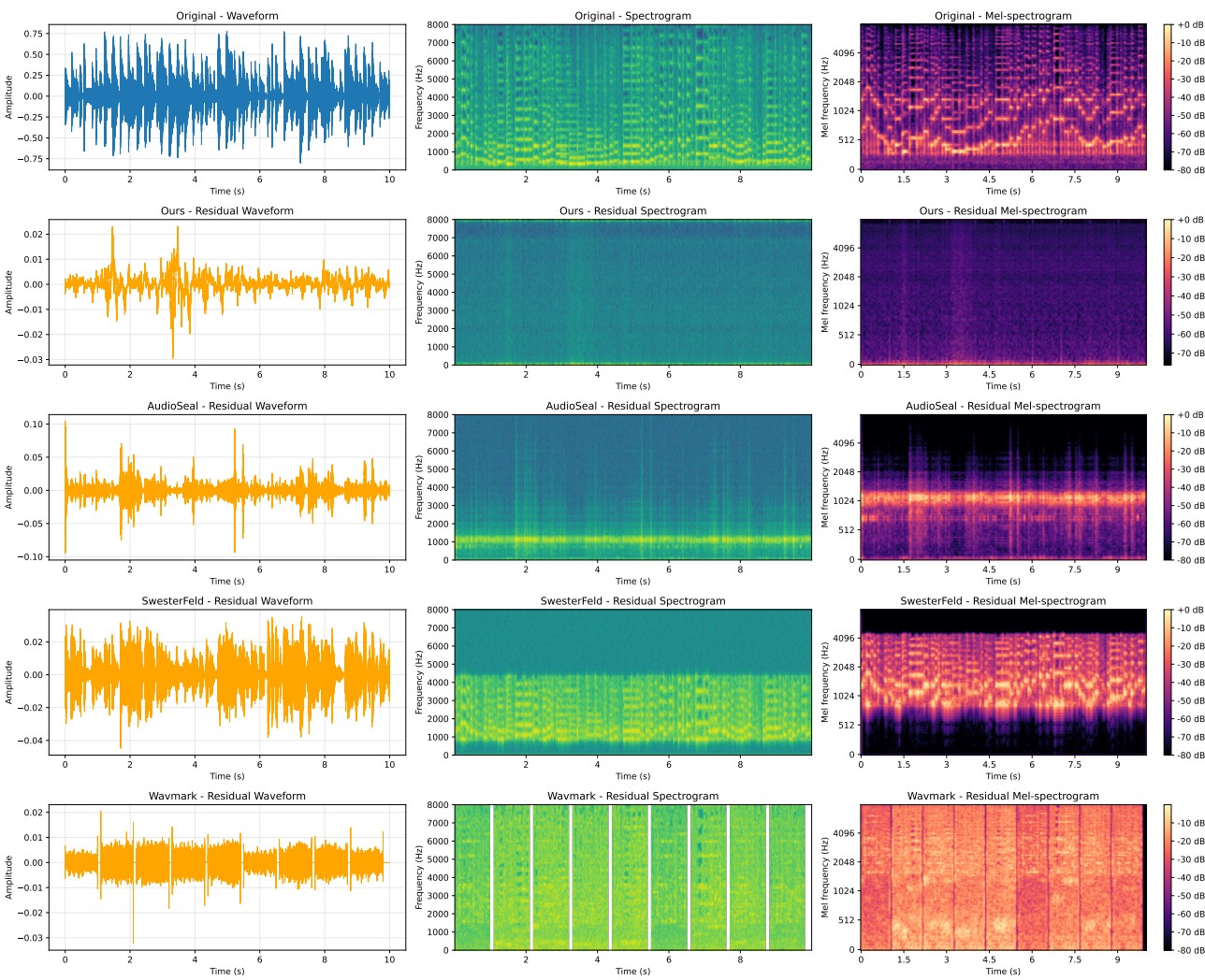

*Figure 11.* Visualization of the original audio and watermark residuals across four different methods for a randomly selected sample from the MusicCaps dataset, showing waveform (left), spectrogram (middle), and mel-spectrogram (right) representations.

where $w$ is the window index and $b$ is the frequency band index. The weighting of the penalty term for each tile is computed in a per-band soft-max manner, $w_b^w = \frac{\exp(l_b^w)}{\sum_{b=1}^B \exp(l_b^w)}$. The final loudness loss is the weighted sum over all the tiles, $\ell_{Loud} = \sum_{w=1}^W \sum_{b=1}^B w_b^w \ell_b^w$.

However, we identify two limitations of this TF-based approach: (1) it lacks consideration of sophisticated auditory masking effects, where the masker and the maskee are not in the same tile; (2) using the loudness difference as the discrepancy measure only provides weak supervision for TF-guided watermarking in the spectrogram domain. To resolve these two issues, we propose to use a more sophisticated TF-weighted $\ell_2$ loss to guide the watermarking process. First, we simulate in the TF domain the 2D energy asymmetric decay. This helps us identifying the potential masker-maskee pairs using filtering rules based on psychoacoustic principles. Secondly, instead of using loudness difference as a discrepancy measure, we compute the mean-square error between the watermarked audio and the original one in the mel-spectrogram domain. This approach provides a denser and finer-grained guidance signal for the imperceptible watermark.

**Visualization of the TF-weighted penalty distribution.** We further visualize the penalty weight distribution of different TF-based approaches of AudioSeal and XATTNMARK with the same spectrogram partitioning configuration. We use a randomly sampled watermarked audio from a model that is at the beginning of training, where the watermark still did not exploit the potential of the TF masking effect for better imperceptibility. As we can see from Figure 12, our method provides

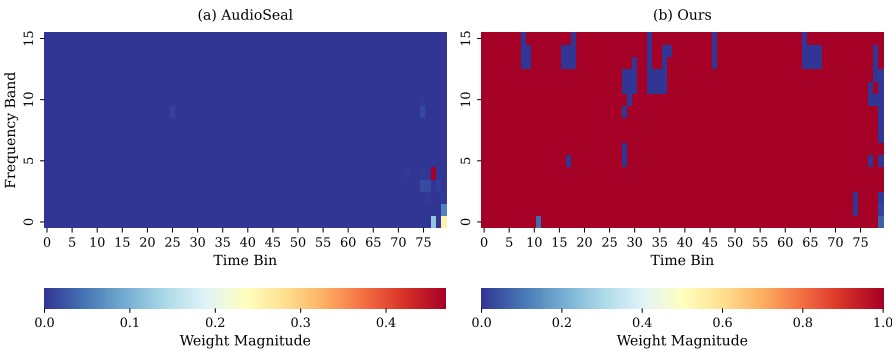

*Figure 12.* Visualization of per-tile masking penalty distributions on the mel-spectrogram. Compared to AudioSeal, our method demonstrates more fine-grained penalty allocation. Blue colors denote regions with lower penalties, preferred for watermark embedding. The audio sample is from MusicCaps (Agostinelli et al., 2023), with the watermark generated by an early-stage model.

a more fine-grained and dense penalty distribution, while AudioSeal's approach just assigns the penalty weight to one tile (the red tile in the figure). Since we use multiple maskers in the vicinity to match their masked regions, our method is able to provide broader and more dense penalty guidance. This indicates that our method is more effective in exploiting the TF masking effect for better imperceptibility.

### C.10. More Results on Watermark Localization

Similar to the Brute Force Detection (BFD) in WavMark (Chen et al., 2023), we further extend our model to have the localization ability. Specifically, since our model includes transformations robust to time shifts and operates on 1s segments, we can distribute the per-segment detection probability to the per-frame level with multiple overlapping detection windows.

Mathematically, let the audio signal be represented as a sequence of frames $F = \{f_1, f_2, \ldots, f_N\}$. Our localization approach utilizes multiple overlapping detection windows, where each window $W_k$ corresponds to a 1s audio segment. For each window $W_k$, our model computes a watermark detection probability $P(M|W_k)$, where $M$ denotes the presence of the watermark. A given frame $f_i$ may be part of several such overlapping windows; let this set be denoted as $\mathcal{W}(f_i) = \{W_k \mid f_i \in W_k\}$. The per-frame detection probability $P(M|f_i)$ is then determined by aggregating the probabilities from all windows covering that frame. We take the average of $P(M|W_k)$ for all $W_k \in \mathcal{W}(f_i)$ as the per-frame detection probability $P(M|f_i)$, i.e., $P(M|f_i) = \frac{1}{|\mathcal{W}(f_i)|} \sum_{W_k \in \mathcal{W}(f_i)} P(M|W_k)$. This process effectively distributes the segment-level detection scores to a finer, per-frame resolution, enabling precise localization of the watermarked regions.

The results are shown in Figure 13. The experiments involve two setups: (1) fixing the audio length to 10s and varying the watermark segment length (2-10s) within it; (2) fixing the watermark segment length to 1s and varying the total audio length (2-10s). For both setups, per-frame level detection is performed. Results show that XATTNMARK can achieve comparable localization performance to AudioSeal and significantly outperforms WavMark.

### C.11. Sensitivity Test in Model Architecture

We investigate the sensitivity of XATTNMARK to key architectural choices, specifically the hidden dimension ($H$) of the message embedding table and the architecture of the temporal conditioning module. The results of this analysis, focusing on convergence speed for message decoding (message-bit validation accuracy), are presented in Figure 14.

For the message embedding table, we explored various hidden dimensions $H$ relative to the secret message bit-length $b$ (fixed at $b = 16$ in our experiments). We performed a grid search over $H \in \{b/2, b, 2b, 4b, 8b\}$. As illustrated in Figure 14a, the choice of $H$ affects the convergence speed of the model, particularly for message decoding. We found that $H$ values ranging from $b/2$ (i.e., 8) to $4b$ (i.e., 64) yield relatively fast convergence. However, using a much larger dimension, such as $H = 8b$ (i.e., 128), can slow down the convergence process (the $H = 128$ case did not converge within 50k training steps).

Regarding the temporal conditioning module, we conducted an ablation study by comparing different MLP architectures: a linear projection (as proposed in XATTNMARK), a 2-layer MLP, and a 3-layer MLP. The results, shown in Figure 14b, indicate that the linear projection is the most effective, being the only one among the tested options that converges

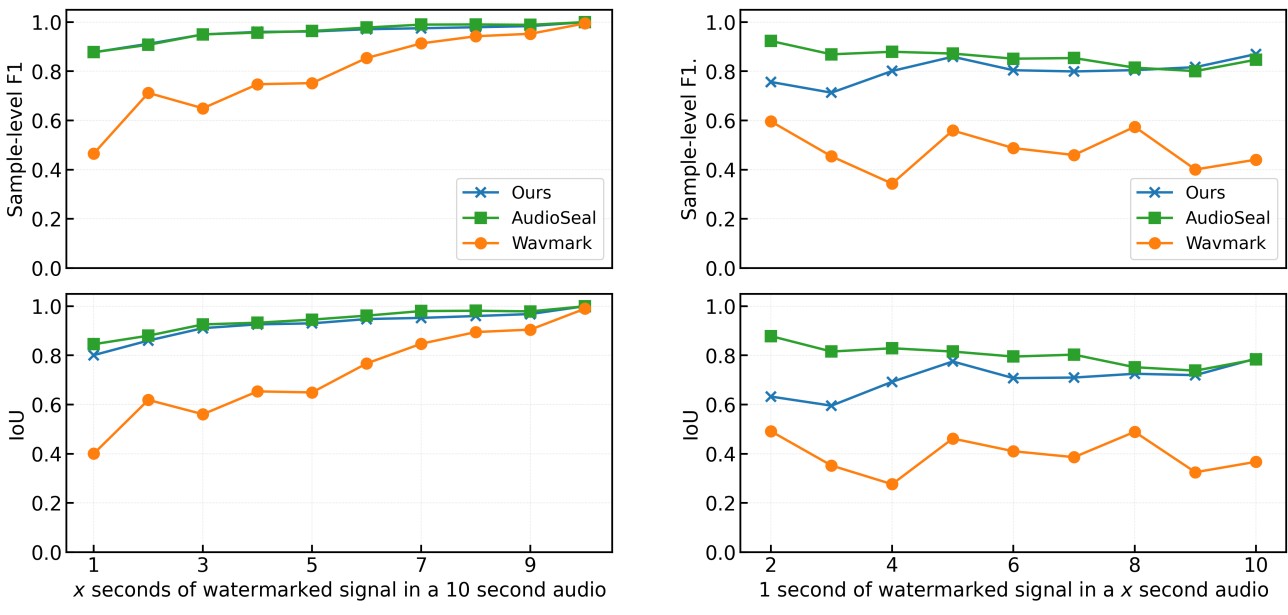

(a) Fixed audio length (10s), varied watermark segment length (2-10s).

(b) Fixed watermark segment length (1s), varied audio length (2-10s).

*Figure 13.* Watermark localization performance comparison. XAttnMark achieves localization comparable to AudioSeal and significantly outperforms WavMark.

successfully within the observed training steps. This suggests that the model's convergence is sensitive to the architectural complexity of the temporal conditioning module.

### C.12. Inference Efficiency and Model Size

We additionally report the model size, computational complexity, training speed, and inference speed comparison with AudioSeal in Table 12. Our model uses fewer parameters and has a smaller size for both the generator and the detector. Although our generator has higher FLOPs and a slightly increased inference time per segment ($\sim$0.3 ms/segment), our detector significantly reduces FLOPs while maintaining similar inference efficiency overall. In training speed, our model achieves a similar second-per-iteration rate (around 1.15 s/iter) as AudioSeal, with faster convergence speed in learning message decoding shown in the App. C.2.

*Table 12.* Comparison of model statistics and inference speed between our method and AudioSeal. Training speed is computed with a batch size of 16 over the first 100 steps.

| Method | Component | #Params ↓ | Model Size ↓ | FLOPs ↓ | Inference Time (ms) ↓ | Train Speed (s/iter) ↓ |
|---|---|---|---|---|---|---|
| Ours | Generator | 10.14M | 38.67 MB | $1.18 \times 10^{11}$ | $0.411 \pm 0.108$ | - |
| | Detector | 7.59M | 28.95 MB | $1.96 \times 10^{11}$ | $0.981 \pm 1.57$ | - |
| | Total | 17.73M | 67.62 MB | $3.14 \times 10^{11}$ | $1.39 \pm 1.58$ | $1.15 \pm 0.51$ |
| AudioSeal | Generator | 14.68M | 56.00 MB | $1.35 \times 10^{10}$ | $0.707 \pm 0.118$ | - |
| | Detector | 8.65M | 32.99 MB | $6.78 \times 10^{11}$ | $0.355 \pm 0.478$ | - |
| | Total | 23.33M | 88.99 MB | $6.92 \times 10^{11}$ | $1.06 \pm 0.493$ | $1.05 \pm 0.27$ |

### C.13. Comparison of Perceptual Quality under Different Watermark Strengths

To compare the perceptual quality of XATTNMARK and AudioSeal under different watermark strengths, we conducted experiments adjusting the watermark intensity. For both XATTNMARK and the AudioSeal baseline, we targeted Signal-to-Noise Ratios (SNRs) at levels from 10 dB to 50 dB, with an interval of 5 dB. The experimental results highlight two

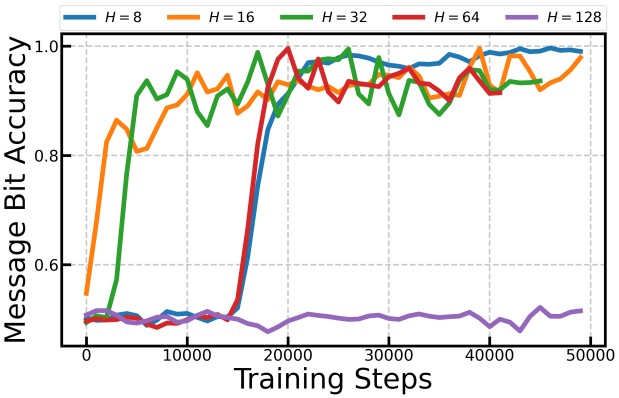

(a) Message-bit validation accuracy with different embedding dimensions ($H$).

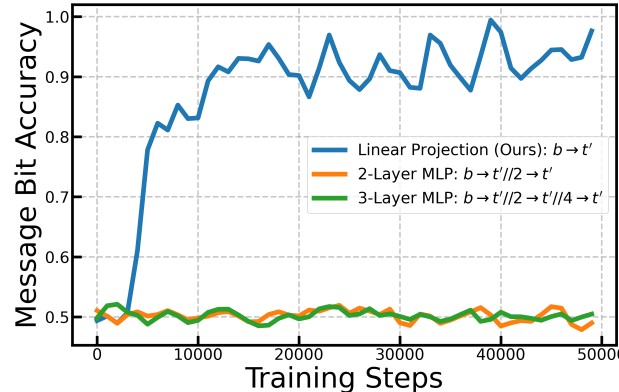

(b) Message-bit validation accuracy with different temporal conditioning architectures.

*Figure 14.* Sensitivity analysis of two model architecture components within 50k training steps. The message-bit validation accuracy is evaluated on the validation set.

key advantages of our approach. First, as illustrated in Figure 15a, the average attribution accuracy (across edits) of XATTNMARK is consistently higher than AudioSeal when compared under different PESQ scores. Second, across different SNR levels, XATTNMARK consistently better or comparable perceptual quality than AudioSeal in terms of PESQ scores. In summary, under different watermark strengths, XATTNMARK achieves better/comparable performance in both attribution accuracy and perceptual quality, compared to AudioSeal.

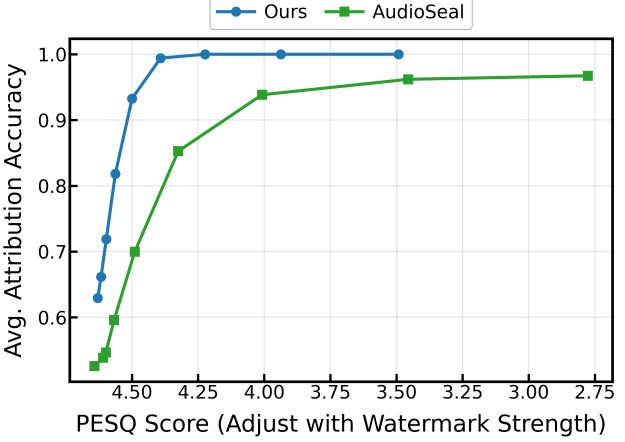

(a) Average attribution accuracy across different PESQ constraints.

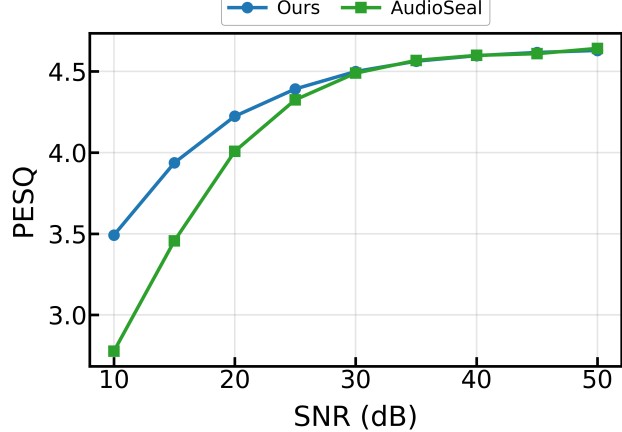

(b) PESQ scores under different SNR constraints.

*Figure 15.* Performance comparison under different perceptual quality constraints with targeted SNR (10-50 dB, 5 dB interval). Figure 15a shows that our method achieves better attribution accuracy across quality levels. Figure 15b shows PESQ under SNR constraints.

### C.14. Statistical Significance Analysis Across Transformations

To rigorously evaluate the performance differences in robustness between XATTNMARK and AudioSeal across various audio transformations, we conduct statistical significance tests. We employ McNemar's test to compare the proportions of correct attributions (a binary outcome for paired samples) and the Wilcoxon signed-rank test to compare the distributions of bit-wise message accuracy scores (a continuous measure for paired samples). In Table 13, we report the results of these tests for each transformation applied to 10k audio samples per transformation. For the vast majority of transformations, XATTNMARK demonstrates statistically significant improvements over AudioSeal. Specifically, the p-values for both McNemar's test (on attribution correctness) and the Wilcoxon signed-rank test (on bit-wise accuracy) are typically far below the standard significance threshold of 0.05 (often $p < 0.001$, indicated as 0.000 in the table). This indicates that the superior performance

of XATTNMARK is not a result of random variation. For instance, in the comprehensive 'Mixed of All' scenario (consisting of all transformations), XATTNMARK achieves an attribution accuracy of 0.88 compared to AudioSeal's 0.58, and a bit-wise accuracy of 0.99 versus 0.93 for AudioSeal. In summary, XATTNMARK achieves statistically significant improvements over AudioSeal across various audio transformations.

*Table 13.* Statistical comparison of watermarking robustness between XAttnMark and AudioSeal on our main setup (user size of $10^4$ is used). For each clip, we generate 100 messages and thus obtain *10k audio samples* in total for each transformation. Results show both McNemar's test on attribution correctness (discrete values) and the Wilcoxon signed-rank test on bit-wise message accuracy (continuous values). Significance is marked as 'Yes' for p-values $< 0.05$ (i.e., $\alpha = 0.05$).

| Transformation | McNemar's Test | | | Wilcoxon Test | | |
|---|---|---|---|---|---|---|
| | **Ours** | **AudioSeal** | **Significant** | **Ours** | **AudioSeal** | **Significant** |
| EnCodec (nq=16) | $0.85{\pm}0.35$ | $0.31{\pm}0.46$ | Yes (0.000) | $0.99{\pm}0.03$ | $0.85{\pm}0.15$ | Yes (9.7e-14) |
| AAC compression | $0.86{\pm}0.34$ | $0.00{\pm}0.02$ | Yes (0.000) | $0.99{\pm}0.03$ | $0.62{\pm}0.10$ | Yes (2.8e-18) |
| Bandpass filter | $0.99{\pm}0.10$ | $0.76{\pm}0.42$ | Yes (0.000) | $1.00{\pm}0.00$ | $0.98{\pm}0.05$ | Yes (7.1e-05) |
| Boost audio | $1.00{\pm}0.02$ | $0.76{\pm}0.43$ | Yes (0.000) | $1.00{\pm}0.00$ | $0.98{\pm}0.04$ | Yes (1.8e-05) |
| Duck audio | $1.00{\pm}0.02$ | $0.78{\pm}0.42$ | Yes (0.000) | $1.00{\pm}0.00$ | $0.98{\pm}0.04$ | Yes (1.2e-05) |
| Echo | $0.98{\pm}0.13$ | $0.50{\pm}0.50$ | Yes (0.000) | $1.00{\pm}0.00$ | $0.93{\pm}0.11$ | Yes (6.8e-10) |
| Highpass filter | $0.99{\pm}0.10$ | $0.77{\pm}0.42$ | Yes (0.000) | $1.00{\pm}0.00$ | $0.98{\pm}0.06$ | Yes (4.4e-05) |
| Identity | $1.00{\pm}0.01$ | $0.78{\pm}0.42$ | Yes (0.000) | $1.00{\pm}0.00$ | $0.98{\pm}0.04$ | Yes (5.9e-05) |
| Lowpass filter | $1.00{\pm}0.02$ | $0.78{\pm}0.42$ | Yes (0.000) | $1.00{\pm}0.00$ | $0.98{\pm}0.04$ | Yes (5.9e-05) |
| MP3 compression | $1.00{\pm}0.03$ | $0.69{\pm}0.46$ | Yes (0.000) | $1.00{\pm}0.00$ | $0.98{\pm}0.05$ | Yes (1.4e-05) |
| Pink noise | $0.99{\pm}0.08$ | $0.77{\pm}0.42$ | Yes (0.000) | $1.00{\pm}0.01$ | $0.98{\pm}0.04$ | Yes (1.7e-05) |
| White noise | $1.00{\pm}0.01$ | $0.76{\pm}0.43$ | Yes (0.000) | $1.00{\pm}0.00$ | $0.98{\pm}0.04$ | Yes (3.6e-05) |
| Smooth | $1.00{\pm}0.02$ | $0.54{\pm}0.50$ | Yes (0.000) | $1.00{\pm}0.00$ | $0.93{\pm}0.10$ | Yes (2.9e-10) |
| Speed | $0.03{\pm}0.17$ | $0.00{\pm}0.05$ | Yes (7.8e-48) | $0.43{\pm}0.08$ | $0.53{\pm}0.14$ | Yes (9.5e-08) |
| Up/down resample | $1.00{\pm}0.02$ | $0.78{\pm}0.42$ | Yes (0.000) | $1.00{\pm}0.00$ | $0.98{\pm}0.04$ | Yes (5.9e-05) |
| Crop | $0.76{\pm}0.43$ | $0.37{\pm}0.48$ | Yes (0.000) | $0.98{\pm}0.06$ | $0.96{\pm}0.09$ | Yes (0.001) |
| **Mixed of All.** | $0.88{\pm}0.33$ | $0.58{\pm}0.49$ | Yes (0.000) | $0.99{\pm}0.07$ | $0.93{\pm}0.13$ | Yes (0.000) |

