# OpenReview forum: "XAttnMark: Learning Robust Audio Watermarking with Cross-Attention"
_ICML.cc/2025/Conference — ICML 2025 poster_

### Official Review · Reviewer_7QuH · 2025-03-11

**Overall Recommendation:** 4

**Summary:**

This paper presents a robust watermarking scheme XAttnMark for audio content, where the embedding and detection of the watermark is performed using neural networks. A key aim of the work is to improve robust attribution (the ability to recover a binary code hidden in the content) while retaining robust detection (the ability to determine whether the content is watermarked or not). The proposed approach builds on AudioSeal, contributing several architectural modifications and a new loss function to improve imperceptibility of the watermark to the human ear. The architectural changes include: (1) sharing of the message conditioning module between the watermark embedding and detection networks, which involves using cross-attention in the detection network; (2) using a learned) linear function for the message conditioning module in place of mean-pooling with temporal-axis repetition. Empirical evaluations indicate that XAttnMark achieves significantly higher attribution accuracy than AudioSeal, with comparable detection accuracy and perceptual quality. XAttnMark is also shown to be the only watermark that achieves reasonably robustness (accuracy > 90%) under editing using audio diffusion models.

## Update after rebuttal

Prior to the rebuttal my main concerns were around:
1. The significance of the empirical results due to small sample sizes
2. Confusion around the cause for the improved performance
3. Unclear motivation for introducing a new perceptual loss

The authors addressed all of these concerns:
1. They explained that the sample size is much larger than 100 as they produce 100 watermarked samples _per_ original audio sample.
2. They corrected my misinterpretation of the ablation study results, reassuring me that parameter sharing is in fact the leading cause for the improved performance.
3. They summarized limitations of AudioSeal's perceptual loss in their response, and noted that this is discussed in an appendix. I encourage to the authors to include a summary of this discussion in the body of the paper.

The authors also provided several new experimental results during the rebuttal period that further enhance the comprehensiveness of the empirical evaluation (evidence of statistical significance, investigation of localized watermarks, more comprehensive attack results, inclusion of false attribute rate for comparison with AudioSeal).

I am now convinced that the paper is sound and will make a strong contribution to the audio watermarking literature. I have therefore increased my score to recommend acceptance.

**Claims And Evidence:**

- The claims of improved robustness and attribution accuracy are based on validation sets of size 100, whereas the AudioSeal paper uses a validation set of size 10,000. For a validation set of size 100, the standard error of accuracy/FPR/TPR could be as large as 5 percentage points, which may call into question the statistical significance of the claims.

- The paper claims that “the fully disjointed architecture of AudioSeal ($\Theta_\mathcal{G} \neq \Theta_\mathcal{D}$) often converges fast for watermark detection learning but struggles to learn the message decoding part efficiently and accurately” (p. 4). However, the ablation study in Fig. 4 suggests that the main limitation may not be the lack of parameter sharing, but rather the choice of message conditioning module. Swapping out the proposed linear message conditioning module with AudioSeal’s results in a message bit accuracy drop from ~98% to ~62%. On the other hand, the use of cross-attention seems to have far less impact on accuracy (between 5-10 percentage points).

- The paper claims that XAttnMask is consistently more robust than AudioSeal against adversarial watermark removal attacks (p. 8). However, the attacks against XAttnMask are generally weaker in terms of their perceptibility than the attacks against AudioSeal, as measured by PESQ, SI-SNR and ViSQOL. Hence the comparison may not be entirely fair. More broadly, the experiments supporting this claim are not as comprehensive as those performed in the AudioSeal paper, which includes stronger gradient-based attacks in the semi-black box and white box settings.

**Essential References Not Discussed:**

I think it’s important to mention the Content Authenticity Initiative (CAI), as an alternative solution to watermarking. It enables tracking of content provenance (including source attribution) via the addition of C2PA metadata, secured by cryptographic means.

**Experimental Designs Or Analyses:**

N/A

**Methods And Evaluation Criteria:**

Yes, the empirical evaluation largely follows norms established in prior work – in terms of datasets, evaluation metrics, and the kinds of benign/adversarial transformations considered for watermark removal.



It’s great to see an ablation study (Table 4 and Figure 4) to assess the impact of the proposed architectural changes in isolation.



My concerns around the evaluation are:

- The use of much smaller validation sets than prior work.

- The definition of attribution accuracy is unclear. In prior work (San Roman et al., 2024), the attribution accuracy is the fraction of examples for which the detection is positive _and_ the attribution is correct. However, in the paper, it appears to be defined as the fraction of detected examples for which the attribution is correct.

- The paper does not report false attribution rate alongside attribution accuracy (see San Roman et al., 2024). This is important as there is a trade-off between false-positives and false-negatives.

- There are no results comparing computational efficiency.

**Other Comments Or Suggestions:**

- Table 3: The names of the quality metrics are introduced in Sec 5.3, after the table is introduced. The columns are missing arrows, indicating whether higher/lower values are better.

**Other Strengths And Weaknesses:**

S1. The empirical evaluation is generally well-executed, apart from my criticism about statistical significance due to small sample sizes. It’s great to see the inclusion of multiple baselines, a range of attacks/benign transformations (including a new diffusion-based attack), and multiple validation datasets. Incidentally, the results for other datasets in Appendix C.6 should be referenced in the body of the paper.



S2. The writing is generally clear. However, I feel the introduction could be made more accessible for readers who are unfamiliar with watermarking and audio.



W1. The new perceptual loss introduced in Section 4.2 is not adequately compared with prior work. I would like to see a qualitative and quantitative comparison with the TF-Loudness loss introduced in AudioSeal. The new loss seems more complicated than TF-Loudness in its construction, so it’s important to provide evidence that the additional complexity has some benefit (e.g., increased detection accuracy for a given level of imperceptibility).



W2. The proposed watermarking scheme does not seem to include _localization_ as a design criterion. In contrast, both WavMark (Chen et al., 2023) and AudioSeal (San Roman et al., 2024) seek to embed _localized_ watermarks in audio, to enable detection of small segments of watermarked audio (e.g., AI-generated speech or copyrighted music) within longer audio clips. By abandoning localization as a constraint, XAttnMark may have an unfair advantage in its ability to achieve high detection/attribution accuracy. I'd like to see some discussion of this in the paper.



W3. A key focus of the paper is on improving source attribution of audio watermarking. However, there is limited discussion explaining why source attribution is important and explaining whether the proposed watermarking scheme addresses the problem. For example, if users of a service are regarded as “sources”, then is a message pool of 10,000 large enough in practice?

**Questions For Authors:**

1. Could the authors comment on the statistical significance of the empirical results.

2. How is attribution accuracy defined? Why are the results for WavMark and AudioSeal different to those reported in the AudioSeal paper?

3. What is the motivation for introducing a new perceptual loss? Is there a problem with the perceptual loss used in AudioSeal?

**Relation To Broader Scientific Literature:**

- Post-hoc neural-network based watermarking. The proposed architecture and training procedure builds on AudioSeal (San Roman et al., 2024) as explained in Appendix A. Similar approaches have been proposed in the image domain – e.g., StegaStamp by Tancik et al. (2020) which is not cited. The claim that AudioSeal “pioneered the disjointed generator-detector paradigm for neural watermarking” (p. 3) is incorrect. StegaStamp is the earliest example I’m aware of, but there may be others.

- New loss for imperceptible watermarks. The proposed loss is inspired by psychoacoustic masking principles (Gelfand, 2017; Holdsworth et al., 1998), recognizing that human listeners struggle to detect small changes in the temporal/frequency proximity of loud sounds. San Roman et al. (2024) also proposed a perceptual loss for audio called TF-Loudness. The paper does not clearly articulate why a new loss is needed, nor how the two losses differ.


**References**

- Tancik, Matthew, Ben Mildenhall, and Ren Ng. "StegaStamp: Invisible hyperlinks in physical photographs." Proceedings of the IEEE/CVF conference on computer vision and pattern recognition. 2020.

**Theoretical Claims:**

N/A

---

> ### Author Rebuttal · Authors · 2025-04-01
>
> We sincerely thank the reviewer for their meticulous and constructive feedback. We will revise the manuscript by improving the introduction and mentioning the CAI initiative and the C2PA standard in the body of the paper. Our responses are as follows:
>
>
> > Q1. On the statistical significance of the results, and concerns on the #validation set.
>
> We want to clarify that, while our test set is indeed composed of 100 audio files, we follow AudioSeal's protocol by hiding 100 messages per file, corresponding to a total of 10k watermarked audio samples. We then apply 16 audio transformations to each one of them. So the total number of samples contributing to our scores is *160k*. To further validate the statistical significance of our results, we perform McNemar's test and Wilcoxon signed-rank tests across edits with 1e4 users. The results are reported in [Tables A and B](https://tinyurl.com/utkjb4yr), which shows that the results on our setup are statistically significant in both attribution performance and perceptual quality.
>
> > Q2. On the definition of attribution accuracy, report of false attribution rate, and the results discrepancy from the two baseline papers.
>
> In our work we define the attribution accuracy in a different way, as the fraction of correct attributions among the *detected* audio inputs, which equals to $1 - FAR$ (False Attribution Rate reported in AudioSeal). This definition decouples the attribution performance from the detection performance. To show a direct comparison between the different metrics, we report the results on both MusicCaps and VoxPopuli setups in [Table C](https://tinyurl.com/3kjjr95f).
>
> > Q3&W.1 On the motivation of the proposed perceptual loss, the comparison with the TF-Loudness loss in AudioSeal, and the justification of additional complexity with score gain.
>
> In Appendix C.8, we discussed the advantage of our proposed masking loss compared to the TF-loudness loss. The TF-loudness loss employs a coarse approach based on loudness differences within each tile, neglecting sophisticated auditory masking effects, such as the interactions between masker and maskee across tiles. Additionally, we found that using loudness difference as a discrepancy measure provides only weak supervision. In contrast, we have designed a more sophisticated TF-weighted MSE loss, which simulates a two-dimensional energy decay in the temporal-frequency domain, effectively identifying masker-maskee pairs, leveraging psychoacoustic principles. Furthermore, we utilize mean-square error in the mel-spectrogram domain as our discrepancy measure, providing more fine-grained guidance (see our qualitative comparison in Figure 10 in the appendix). To quantitatively justify the effectiveness of our proposed loss, we evaluate the attribution accuracy under different watermark strengths (controlled by PESQ ranges) in [Figure A](https://tinyurl.com/y38skm78). These results clearly indicate that our method consistently achieves significantly higher attribution accuracy at each imperceptibility level.
>
>
> > Q4. On the lack of results comparing computational efficiency.
>
> We additionally report the results on computational efficiency in [Table D](https://tinyurl.com/2p9p5cpd).
>
> > W2. On the localization capabilities of XAttnMark.
>
> Thanks for this insightful point. Although we have not explicitly discussed the localization capabilities of XAttnMark, our model can be easily extended to have this ability with sliding window detection. Specifically, since our model includes shifting-robust transformations and operates on 1s segments, we can distribute the per-segment detection probability to the per-frame level with multiple overlapping detection windows as the BFD in WavMark does. We have implemented this and report the results in [Figure B](https://tinyurl.com/yxtft2mb). Results show that XAttnMark can achieve comparable localization performance to AudioSeal and significantly outperforms WavMark.
>
>
> > Q5. On the comparison with stronger gradient-based attacks in the semi-black box and white box settings and fairness on the HSJA.
>
> We additionally report the robustness against the white-box, semi-black-box attacks in [Figure C](https://tinyurl.com/2xmc3km3). Results show that XAttnMark is slightly more vulnerable to these two attacks compared to AudioSeal (might be attributed to our smaller detector). On the fairness concern on HSJA, we clarify that we use the same attack budget for the two methods, and the higher perceptibility score in ours is because HSJA fails to successfully find more adversarial samples within the given budget compared to AudioSeal's case.
>
> > Q6. On the contribution of the cross-attention and the conditioning module.
>
> Regarding the interpretation of Fig. 4, we clarify that our claim is that, when considering a single module in isolation, the cross-attention module (acc. of 62%) is more effective than the temporal conditioning module (acc. of 50%) in enhancing efficiency.

---

> > ### Comment · Reviewer_7QuH · 2025-04-02
> >
> > I appreciate your detailed rebuttal. The new empirical results you've shared will round out the paper nicely. I'm satisfied with the responses to my concerns, and will update my score accordingly.

---

> > > ### Author Response · Authors · 2025-04-04
> > >
> > > We sincerely thank the reviewer for their careful consideration and updated assessment. We are pleased that our response has addressed your concerns. We will incorporate the additional results into the revised version of the manuscript accordingly.

---

### Official Review · Reviewer_f1fs · 2025-03-13

**Overall Recommendation:** 3

**Summary:**

This paper focuses on robust audio watermark detection and source attribution, which is more a technical report than a top-tier conference paper. Specifically, it adopts blended architecture of disjointed generator-detector and fully shared-parameter architecture. Besides, temporal conditioning mechanism and per-tile temporal-frequency masking loss are utilized to improve watermarking performance. In general, the writing is good from the technical aspect. It emphasizes the details of each contribution, lacking more thorough and deep analysis. The experimental results show its effectiveness of proposed method.

**Claims And Evidence:**

The experimental results, including comparison with SOTA and ablation studies show its effectiveness of proposed method.

**Essential References Not Discussed:**

The references seem adequate.

**Experimental Designs Or Analyses:**

The experimental results, including comparison with SOTA and ablation studies show its effectiveness of proposed method.

**Methods And Evaluation Criteria:**

As discussed above, the paper focus on illustrating technical details while lacking more deeper insights for robust audio watermarking.

**Other Comments Or Suggestions:**

The writing is good in general. However, in my opinion, it is more like a technical report.

**Other Strengths And Weaknesses:**

As discussed above, the motivation behind the key contributions is not clear, which merely introduce problems and propose detailed technique and verify it with experiments. Although most papers are similar style with it, this paper is much more to be a technical report than other papers to be reviewed.

**Questions For Authors:**

How about the comparison of model's parameters, training speed and inference speed?

**Relation To Broader Scientific Literature:**

It achieves both better performance on audio watermark detection and source attribution compared with SOTA methods.

**Theoretical Claims:**

There are no proofs for theoretical claims.

---

> ### Author Rebuttal · Authors · 2025-04-01
>
> We sincerely appreciate the reviewer's insightful attention and precise feedback. We have addressed the reviewer's concerns as follows:
>
> > Q1: How about the comparison of model's parameters, training speed and inference speed?
>
> **Response:** We additionally report the model size, training speed, and inference speed comparison with AudioSeal in [Table D](https://tinyurl.com/2p9p5cpd). Our model uses fewer parameters and has a smaller size for both the generator and the detector. Although our generator has higher FLOPs and a slightly increased inference time per segment (~0.3 ms/segment), our detector significantly reduces FLOPs while maintaining similar inference efficiency overall. In training speed, our model achieves a similar second-per-iteration rate (around 1.15s/iter) as AudioSeal, with faster convergence speed in learning message decoding. Specifically, as shown in Appendix C.2, our model takes ~4k steps to reach perfect detection accuracy, and ~10k steps to reach perfect attribution accuracy, while AudioSeal takes ~32k steps to reach perfect detection accuracy, and 50k steps to reach around 70% attribution accuracy. This demonstrates that XAttnMark achieves 5 to 8 times better training efficiency than AudioSeal.
>
> ---
>
> > Q2: As discussed above, the paper focuses on illustrating technical details while lacking more deeper insights for robust audio watermarking.
>
> **Response:** Thank you for this valuable point. We will refine our presentation to include more design insights. Due to the space limit, we put a significant part of the details on the design motivation in the appendix sections. In the appendix, we have provided more analysis and discussion on the proposed modules, including the cross-attention architecture and the proposed temporal-frequency perceptual loss. Specifically,
> - In Appendix C.2. (Analysis of the Training Dynamics of Models with Different Architectures), we analyze the training dynamics of different architectures under a controlled experimental setup to better understand their inherent learning capabilities.
> - In Appendix C.8, we provide a comprehensive comparison with the TF-loudness loss of AudioSeal.
>
> During the revision of the paper, we will add these deeper technical insights to the main text for better readability.

---

### Official Review · Reviewer_otVN · 2025-03-14

**Overall Recommendation:** 3

**Summary:**

The paper introduces a novel neural audio watermarking framework called XATTNMARK. The key contributions include:
A cross-attention mechanism that enables efficient message retrieval by sharing an embedding table between the generator and detector.
A temporal conditioning module that distributes the message temporally, improving learning efficiency.
A psychoacoustic-aligned temporal-frequency masking loss that enhances watermark imperceptibility by leveraging human auditory masking effects.
The main findings show that XATTNMARK achieves state-of-the-art performance in both detection and attribution, demonstrating superior robustness against a wide range of audio transformations, including challenging generative editing.

**Claims And Evidence:**

The claims made in the paper are well-supported by clear and convincing evidence.

**Essential References Not Discussed:**

No

**Experimental Designs Or Analyses:**

The experimental designs and analyses in the paper are generally sound and provide strong evidence to support the claims.
The subjective listening test involves a relatively small number of participants.
The ablation study on the adaptive bandwidth (constant weight γ = 1) is limited to a single configuration.

**Methods And Evaluation Criteria:**

The evaluation on diverse audio transformations and benchmark datasets provides a rigorous assessment of the method's robustness and practical applicability.

**Other Comments Or Suggestions:**

No

**Other Strengths And Weaknesses:**

No

**Questions For Authors:**

See above

**Relation To Broader Scientific Literature:**

The key contributions of the paper are well-grounded in the broader scientific literature on audio watermarking and generative audio technologies. XATTNMARK builds upon previous work by introducing innovative mechanisms for message retrieval, temporal conditioning, and psychoacoustic alignment.

**Theoretical Claims:**

The theoretical claims in the paper are supported by empirical evidence and are grounded in well-established principles.
The paper mentions adversarial attacks but does not provide detailed theoretical analysis on the robustness of XATTNMARK against such attacks.

---

> ### Author Rebuttal · Authors · 2025-04-01
>
> We sincerely thank the reviewer for taking the time to review our manuscript and providing valuable feedback. We have carefully considered each point raised and provide our detailed responses below:
>
> > Q1. The subjective listening test involves a relatively small number of participants.
>
> **Response:** We launch our subjective listening test with 18 participants initially following the ITU-R BS1534-1 [1] standard and practice in related audio publications [2,3]. Specifically, ITU-R BS1534-1 suggests that **when the conditions of a listening test are tightly controlled on both the technical and behavioral side, experience has shown that data from no more than 20 subjects are often sufficient for drawing appropriate conclusions from the test**. Our internal test with unified software/process and participants from a similar background profile satisfies this control requirement.
>
> During the post-screening process, we further filtered out 6 participants who missed the reference audio to ensure the result's validity, resulting in 12 valid evaluators. Similarly, SilentCipher[3] also performs post-processing on the test group results with 12 valid evaluators in total. While these references support our setup, we acknowledge that the population size for our MUSHRA test is relatively limited. If needed, we will expand the test population and update our results in the final version of the paper.
>
> [1] Method for the subjective assessment of intermediate quality level of coding systems (Recommendation ITU-R BS.1534-1), International Telecommunication Union. (2003).
>
> [2] Davidson, G., Vinton, M., Ekstrand, P., Zhou, C., Villemoes, L., & Lu, L. (2023). High Quality Audio Coding with MDCTNet. IEEE International Conference on Acoustics, Speech, and Signal Processing (ICASSP).
>
> [3] Singh, M. K., Takahashi, N., Liao, W., & Mitsufuji, Y. (2024). SilentCipher: Deep Audio Watermarking. In Proc. Interspeech 2024 (pp. 2235-2239).
>
> ---
>
> > Q2. The ablation study on the adaptive bandwidth (constant weight $\gamma$ = 1) is limited to a single configuration.
>
> **Response:** Our ablation study on the adaptive bandwidth mainly focuses on the two most representative cases, the constant weighting and the adaptive weighting for per-mel masking radii, which we propose in our work. In our design, the per-mel-bin masking radii $r^m$ are adjusted based on the frequency instead of being set as a constant across all frequencies. Adjusting different $\gamma$ values can be viewed as a hyperparameter tuning process on the base radius $r^m_b$, which still assigns the same radii across all the mel-bins and conceptually belongs to the same class of constant weighting. Due to the time constraint, we leave the exploration on this hyperparameter searching for constant weighting as a future work.

---

### Official Review · Reviewer_fJ8F · 2025-03-24

**Overall Recommendation:** 3

**Summary:**

This paper proposes XATTNMARK, a novel neural audio watermarking system designed to achieve both robust detection and accurate message attribution, two goals that are difficult to achieve simultaneously in prior work. The authors blend the architectural benefits of WavMark and AudioSeal by introducing partial parameter sharing between the generator and detector, enabled via a cross-attention decoding mechanism and a shared embedding table. Additionally, a temporal message conditioning module and a psychoacoustic-aligned time-frequency masking loss are proposed to enhance imperceptibility and robustness. Experiments demonstrate that XATTNMARK achieves SOTA performance across a wide range of audio transformations, including generative edits, and adversarial attacks, while maintaining high perceptual quality.

**Claims And Evidence:**

The central claim is that XATTNMARK simultaneously achieves robust detection and accurate attribution across diverse audio transformations, outperforming existing SOTA methods. The empirical evidence, especially Table 1 and Table 2, supports this claim convincingly. The robustness against generative editing and adversarial removal is particularly noteworthy, as prior methods degrade significantly under such settings. Ablation studies (Figure 4) and quality assessments (Table 4) further strengthen the evidence for each architectural component’s contribution.

**Essential References Not Discussed:**

None glaring.

**Experimental Designs Or Analyses:**

The paper extensively benchmarks detection and attribution across a wide range of realistic scenarios, including speed edits, generative model edits, and adversarial attacks. The performance gains are statistically significant, and trade-offs are clearly analyzed. Ablations are especially useful in isolating the impact of core contributions.

**Methods And Evaluation Criteria:**

The methodology is technically sound. The partial parameter sharing via a shared embedding table and cross-attention decoding is well-motivated and novel. The experimental protocol is thorough, using 16 types of transformations, two generative editing models (AudioLDM2, Stable Audio), and adversarial perturbations (HSJA). Baselines are strong (AudiowMark, WavMark, TimbreWM, AudioSeal), and evaluation metrics include detection/attribution accuracy, perceptual audio quality, and robustness under various threats.

**Other Comments Or Suggestions:**

- The paper would benefit from clarifying the decoding pipeline under attribution evaluation with large user pools (e.g., scalability of Hamming decoding).

- Consider releasing code/models to improve reproducibility and adoption.

**Other Strengths And Weaknesses:**

Strengths:

- Strong empirical gains across detection and attribution.

- Well-structured methodology with insightful architectural design.

- Broad experimental coverage (standard, generative, adversarial).

- High practical relevance in the age of generative audio content.

Weaknesses:

- The model still struggles with extreme transformations like speed changes (acknowledged in the text).

- Attribution performance under generative edits was not deeply analyzed; only detection is reported.

- The paper does not evaluate robustness under white-box adversarial attacks.

**Questions For Authors:**

- Can the attribution method be extended to a continuous space (e.g., using embeddings) to improve robustness under generative edits?

- How sensitive is the performance to the architecture of the embedding table and temporal conditioning module?

- The paper does not evaluate robustness under white-box adversarial attacks. How would the proposed method perform under white-box adversarial attacks compared with existing methods?

**Relation To Broader Scientific Literature:**

The paper properly contextualizes its contribution within prior watermarking methods (AudiowMark, WavMark, AudioSeal), as well as broader work on dataset attribution and copyright auditing. References are recent and well-curated.

**Theoretical Claims:**

The paper is mostly empirical, but the formulation of the psychoacoustic-aligned temporal-frequency masking loss is theoretically grounded in auditory perception literature. The architecture and cross-attention design are sound from a deep learning perspective.

---

> ### Author Rebuttal · Authors · 2025-04-01
>
> We sincerely thank the reviewer for their valuable feedback. We have addressed the reviewer's concerns as follows:
>
> > W1. The model still struggles with extreme transformations like speed changes (acknowledged in the text).
>
> In the paper we show that the model is able to effectively perform the detection task under speed changes transformations. We also acknowledge that the model struggles against speed changes (and other challenging transformations like generative edits) for the attribution task. However, in the Appendix C.3.1, we show that, for challenging speeding operation, we could build up a simple speed reversion layer, that greatly improve the attribution performance without significant overhead (as shown in Table 7 in appendix).
>
> > W2. Attribution performance under generative edits was not deeply analyzed; only detection was reported.
>
> As mentioned earlier, we acknowledge that our model is still limited in attribution robustness against generative edits. However, we believe that this still marks a significant step forward in watermarking against generative audio edits. To the best of our knowledge, we are the first to report non-trivial detection robustness (90%+) against generative editing in a zero-shot manner. With additional specialized training on those transformations, the attribution performance might be further improved. We leave this as future work.
>
> > W3. The paper does not evaluate robustness under white-box adversarial attacks.
>
> We additionally report the robustness against white-box adversarial attacks in [Figure C](https://tinyurl.com/2xmc3km3).
>
>
> > C1. The paper would benefit from clarifying the decoding pipeline under attribution evaluation with large user pools (e.g., scalability of Hamming decoding).
>
> Please refer to the 'Evaluation Setup' part of Appendix A and Table 6, where we have provided a detailed discussion on the Hamming decoding process used in the attribution evaluation and also the scalability aspect.
>
> > C2. Consider releasing code/models to improve reproducibility and adoption.
>
> Thank you for the suggestion. We will consider releasing the code and models upon the paper's publication.
>
> > Q1: Can the attribution method be extended to a continuous space (e.g., using embeddings) to improve robustness under generative edits?
>
> This is an interesting idea. Currently, the existing watermarking methods are mostly designed for embedding discrete bit-strings (e.g., 0s and 1s). However, our experiments show that, under the challenging generative editing, previous methods fail to succeed at both detection and attribution. One potential reason is that we treat the source as discrete informationless bit-strings, without leveraging the semantic information that could help the attribution task (e.g., style attribution [1]). For example, in the audio domain, a copyrighted timbre might have countless reference audio files, which could be leveraged as attribution anchors for more robust attribution. This is an orthogonal direction to our research, which we leave as future work.
>
> [1] Wang, Sheng-Yu, et al. "Evaluating data attribution for text-to-image models." Proceedings of the IEEE/CVF International Conference on Computer Vision. 2023.
>
> > Q2: How sensitive is the performance to the architecture of the embedding table and temporal conditioning module?
>
> We additionally provide the sensitivity study on the embedding hidden dimension and temporal conditioning architecture in [Figure D](https://tinyurl.com/49xfer89). For the embedding table, we found that the embedding dimension $H$ will affect the convergence speed of the watermark model in both detection and message decoding (more sensitive to $H$ in the message decoding part). With grid search over $[\frac{b}{2}, b, 2b, 4b, 8b]$, where $b$ is the bit-length of the secret message ($b=16$), we found that $H$ values ranging from $H=\frac{b}{2}$ to $H=4b$ both yield fast convergence except for $H=8b$. For the temporal conditioning module, we additionally provide an ablation study with different numbers of MLP layers (linear, 2-layer MLP, and 3-layer MLP). Results show that the linear projection proposed in XAttnMark is the only one that converges, indicating that the convergence is sensitive to the architecture choice of the temporal conditioning module.
>
> > Q3: The paper does not evaluate robustness under white-box adversarial attacks. How would the proposed method perform under white-box adversarial attacks compared with existing methods?
>
> We additionally report the robustness against white-box, semi-black-box, and Gaussian noise attacks in [Figure C](https://tinyurl.com/2xmc3km3). The results show that XAttnMark is more robust than AudioSeal in Gaussian noise attacks. In the white-box and semi-black-box attack scenario, we observe that XAttnMark is slightly more vulnerable to white-box attacks compared to AudioSeal, which might be due to the smaller model size of the detector module (XAttnMark is 7.59M, while AudioSeal is 8.65M).

---

### Decision · Program_Chairs · 2025-05-01

**Decision:**

Accept (poster)

**Comment:**

This paper presents XATTNMARK, a neural audio watermarking framework that aims to deliver both robust watermark detection and precise message attribution. The proposed approach combines elements from prior work (i.e., WavMark and AudioSeal), incorporating partial parameter sharing between the generator and detector through a cross-attention decoding mechanism and a shared embedding table.

Reviewers found the method compelling, with strong performance and clear presentation. However, they also pointed out a lack of detail regarding the method’s motivation and noted some missing evaluations. The authors addressed most of these concerns with additional results and methodological clarifications. I encourage the authors to include these additional experiments as part of their manuscript, this will greatly improve their submission.